# In vivo rendezvous of small nucleic acid drugs with charge-matched block catiomers to target cancers

Sumiyo Watanabe[1,2,3,13], Kotaro Hayashi[4,13], Kazuko Toh[4,13], Hyun Jin Kim[1,13], Xueying Liu[4], Hiroyuki Chaya[1,5], Shigeto Fukushima[4], Keisuke Katsushima[6], Yutaka Kondo[6], Satoshi Uchida[7], Satomi Ogura[4,5], Takahiro Nomoto[8], Hiroyasu Takemoto[8], Horacio Cabral[7], Hiroaki Kinoh[4], Hiroyoshi Y. Tanaka[9], Mitsunobu R. Kano[9,10], Yu Matsumoto[1], Hiroshi Fukuhara[11], Shunya Uchida[2], Masaomi Nangaku[3], Kensuke Osada[7], Nobuhiro Nishiyama[8], Kanjiro Miyata[1,5,14] & Kazunori Kataoka[4,12,14]

Stabilisation of fragile oligonucleotides, typically small interfering RNA (siRNA), is one of the most critical issues for oligonucleotide therapeutics. Many previous studies encapsulated oligonucleotides into ~100-nm nanoparticles. However, such nanoparticles inevitably accumulate in liver and spleen. Further, some intractable cancers, e.g., tumours in pancreas and brain, have inherent barrier characteristics preventing the penetration of such nanoparticles into tumour microenvironments. Herein, we report an alternative approach to cancer-targeted oligonucleotide delivery using a Y-shaped block catiomer (YBC) with precisely regulated chain length. Notably, the number of positive charges in YBC is adjusted to match that of negative charges in each oligonucleotide strand (i.e., 20). The YBC rendezvouses with a single oligonucleotide in the bloodstream to generate a dynamic ion-pair, termed unit polyion complex (uPIC). Owing to both significant longevity in the bloodstream and appreciably small size (~18 nm), the uPIC efficiently delivers oligonucleotides into pancreatic tumour and brain tumour models, exerting significant antitumour activity.

[1] Center for Disease Biology and Integrative Medicine, Graduate School of Medicine, The University of Tokyo, 7-3-1 Hongo, Bunkyo-ku, Tokyo 113-0033, Japan. [2] Division of Nephrology, Department of Internal Medicine, Teikyo University School of Medicine, 2-11-1 Kaga, Itabashi-ku, Tokyo 173-8605, Japan. [3] Division of Nephrology and Endocrinology, Graduate School of Medicine, The University of Tokyo, 7-3-1 Hongo, Bunkyo-ku, Tokyo 113-8544, Japan. [4] Innovation Center of NanoMedicine, Kawasaki Institute of Industrial Promotion, 3-25-14 Tonomachi, Kawasaki-ku, Kawasaki 210-0821, Japan. [5] Department of Materials Engineering, Graduate School of Engineering, The University of Tokyo, 7-3-1 Hongo, Bunkyo-ku, Tokyo 113-8656, Japan. [6] Division of Cancer Biology, Nagoya University Graduate School of Medicine, 65 Tsurumai-cho, Showa-ku, Nagoya 466-8550, Japan. [7] Department of Bioengineering, Graduate School of Engineering, The University of Tokyo, 7-3-1 Hongo, Bunkyo-ku, Tokyo 113-8656, Japan. [8] Laboratory for Chemistry and Life Science, Institute of Innovative Research, Tokyo Institute of Technology, R1-11, 4259 Nagatsuta, Midori-ku, Yokohama 226-8503, Japan. [9] Department of Pharmaceutical Biomedicine, Okayama University Graduate School of Medicine, Dentistry, and Pharmaceutical Sciences, 1-1-1 Tsushima-naka, Kita-ku, Okayama-shi, Okayama Prefecture 700-8530, Japan. [10] Department of Pharmaceutical Biomedicine, Okayama University Graduate School of Interdisciplinary Science and Engineering in Health Systems, 1-1-1 Tsushima-naka, Kita-ku, Okayama-shi, Okayama Prefecture 700-8530, Japan. [11] Department of Urology, Kyorin University Faculty of Medicine, 6-20-2 Shinkawa, Mitaka, Tokyo 181-8611, Japan. [12] Institute for Future Initiatives, The University of Tokyo, 7-3-1 Hongo, Bunkyo-ku, Tokyo 113-0033, Japan. [13] These authors contributed equally: Sumiyo Watanabe, Kotaro Hayashi, Kazuko Toh, Hyun Jin Kim. [14] These authors jointly supervised this work: Kanjiro Miyata, Kazunori Kataoka. Correspondence and requests for materials should be addressed to K.M. (email: miyata@bmw.t.u-tokyo.ac.jp) or to K.K. (email: kataoka@ifi.u-tokyo.ac.jp)

Transporting small nucleic acids (SNAs), such as small interfering RNA (siRNA) and antisense oligonucleotide (ASO), to a target tissue by the systemic route remains an open issue for controlling specific gene expression with therapeutic efficacy[1–3]. The stabilisation of fragile SNAs under harsh in vivo conditions is one of the most critical issues. Many previous studies have focused on the encapsulation of SNAs into ~100 nm-sized nanoparticles through multimolecular assembly. However, such nanoparticles accumulate to a considerable extent in the reticuloendothelial system of the liver and spleen. Furthermore, some intractable cancers have inherent barrier characteristics—e.g. the thick fibrotic stromal tissue surrounding cancer cell nests in pancreatic cancer and the blood–brain tumour barrier in brain tumours—that prevent the penetration of such nanoparticles into the tumour microenvironments[4–8,9].

To overcome the above issue, the present study establishes an alternative approach to SNA stabilisation in vivo by engineering a Y-shaped block catiomer (YBC) with precisely regulated chain length, which abrogates the need for large nanoparticles. Indeed, the number of positive charges in the YBC is adjusted to match the number of negative charges in each SNA strand (i.e. 20). This charge-matching enables dynamic and selective ion-pairing of a single molecule of siRNA or ASO with a dyad or single molecule(s) of YBC in the bloodstream, generating a dynamically equilibrated unit polyion complex (uPIC). Both siRNA and ASO circulate stably in the bloodstream in a scheme of autonomous rendezvous with free YBCs. Owing to their appreciably small size comparable to antibody, the uPICs are able to enter both stroma-rich pancreatic tumours and brain tumour tissues. Ultimately, the uPICs exert significant antitumour activity for a spontaneous pancreatic tumour model and an orthotopic brain tumour model by transporting an apoptosis-inducing siRNA coding *polo-like kinase 1* (*PLK1*) and an ASO coding *turine upregulated gene 1* (*TUG1*), respectively.

## Results

**Fabrication and characterization of uPIC**. We first demonstrated the spontaneous formation of uPIC in an aqueous solution using siRNA and YBC, a block catiomer comprising poly(L-lysine) (PLL) and two-armed poly(ethylene glycol) (PEG) (molecular weight (MW) of PEG = $2 \times 37$ kDa)[10,11]. To selectively fabricate a uPIC composed of a single siRNA and dyad YBCs with a tetrahedral structure (Fig. 1a), it was important to restrict the number of cationic amines ($N_A$) in the PLL segment close to 20, consistent with the number of anionic phosphates ($N_P$) in each strand of the siRNA (21-mer/21-mer), with a narrow MW distribution ($M_w/M_n < 1.1$) (Supplementary Fig. 1). The formation of a charge-neutralised PIC from siRNA (GL3 luciferase-targeted siRNA, siLuc) and YBC was confirmed by agarose gel electrophoresis (Fig. 1b) and electrophoretic laser scattering (Fig. 1c). On the electrophoresed gel, the naked siRNA band disappeared at stoichiometric and higher charge ratios, i.e. A/P: [residual amino groups in YBC]/[residual phosphate groups in siRNA] $\geq 1$. Moreover, the negative zeta potential of the siRNAs was completely neutralised at A/P $\geq 1$. We then verified single siRNA-loaded PIC (siRNA/uPIC) formation at A/P $\geq 1$ by fluorescence correlation spectroscopy (FCS) using Alexa Fluor® 647-labelled siRNA (A647-siRNA) (Fig. 1d). There was a linear increase in the average hydrodynamic diameter ($D_H$) of A647-siRNA with an increasing A/P ratio until the charge neutralisation point ($0 < $ A/P $ < 1$), indicating the generation of PICs by charge-stoichiometric interactions between siRNAs and YBCs. The $D_H$ of the PICs plateaued at 18 nm after reaching the charge neutralisation point (A/P $\geq 1$), suggesting that excess YBCs over charge stoichiometry may remain in free form. The average

amplitude of the fluorescence fluctuations (or amplitude per particle) in the FCS measurements was the same for the PIC samples as for the free A647-siRNA, regardless of the A/P ratio (Fig. 1d). This result indicates that only a single siRNA was incorporated into each PIC.

The structure of the uPIC was further investigated using a sedimentation equilibrium method in an analytical ultracentrifuge (AUC) to determine the MW. The MW of the uPIC prepared at A/P = 1 was $150 \pm 9$ kDa (mean ± s.d. measured three times), which is comparable to that predicted for the 2:1 uPIC composed of dyad YBCs (MW = 76 kDa for each) and a single siRNA (MW = 13 kDa). Furthermore, the MW distribution of the uPICs, determined by a sedimentation velocity method in an AUC, displayed a unimodal sharp peak similar to that of free YBC (Fig. 1e), indicating monodispersed uPIC formation. Generally, PIC formation between block catiomers and aniomers is followed by the multimolecular secondary association, generating micellar (or vesicular) PICs (Supplementary Fig. 2)[10–12]. The selective fabrication of uPICs without the formation of micellar PICs is quite unusual. Presumably, the steric repulsive effect of the two-armed PEG surrounding the PLL/siRNA ion-pair and the siRNA rigidity arising from the double-stranded architecture prevent the secondary associations of uPICs[11,13]. It should be noted that the MW of the uPIC remained constant at A/P > 1 (Supplementary Fig. 3), consistent with the FCS result, revealing that excess YBCs over A/P = 1 exist in free form in solution.

**Evaluation of dynamic nature of uPIC**. The dynamic equilibrium between the uPICs and the free YBCs, illustrated in Supplementary Movie 1 and Supplementary Figure 4, was verified by monitoring the exchange reaction between the uPIC-forming YBCs and the free YBCs using fluorescence resonance energy transfer (FRET) (Fig. 2a). Alexa Fluor® 594-labelled YBC (A594-YBC) was mixed with A647-siRNA at A/P = 1 to form FRET-uPIC, which exhibited a high FRET signal (defined as the ratio of the fluorescence intensity of A647 to the fluorescence intensity of A594 under excitation at 561 nm) due to the proximate dyes within uPIC (Fig. 2b). The FRET signal was significantly reduced by the addition of non-labelled YBC (N-YBC) (final A/P = 2) (Fig. 2b), indicating that an exchange reaction occurred between A594-YBC and N-YBC, which affected the FRET signal. The dynamic equilibrium between the uPICs and the YBCs in the bloodstream was ultimately evidenced using a similar FRET methodology combined with intravital confocal laser scanning microscopy (IVCLSM) (Supplementary Movie 2). A594-YBC (20 nmol) was intravenously injected into a mouse, followed 4 min later by intravenous injection of A647-siRNA (2 nmol). The FRET signal from the vein increased visibly following the injection of siRNA (Fig. 2c, d, f). In contrast, an additional intravenous injection of N-YBC (20 nmol) resulted in a reduction in the FRET signal (Fig. 2e, f), similar to the FRET analysis performed in the buffer solution. Thus, the dynamic and selective ion-pairing, namely rendezvous, between the siRNA and YBCs demonstrably persisted in the bloodstream containing various charged biomacromolecules, which may act as potential competitors.

To verify that the PICs formed in the bloodstream comprised dyad YBCs and a single siRNA (i.e. 2:1 uPIC), we further analysed the structure of PICs harvested from the bloodstream by FCS and AUC. As shown in Fig. 2g, three uPIC samples were prepared from A647-siRNA and YBC (or N-YBC) for comparison: sample (i) was prepared at A/P = 10 by simply mixing YBC (40 nmol) with A647-siRNA (2 nmol) in the buffer solution without administration; sample (ii) was similarly prepared in the buffer solution, intravenously injected into a mouse, and harvested as a supernatant fluid of the blood (or plasma); and

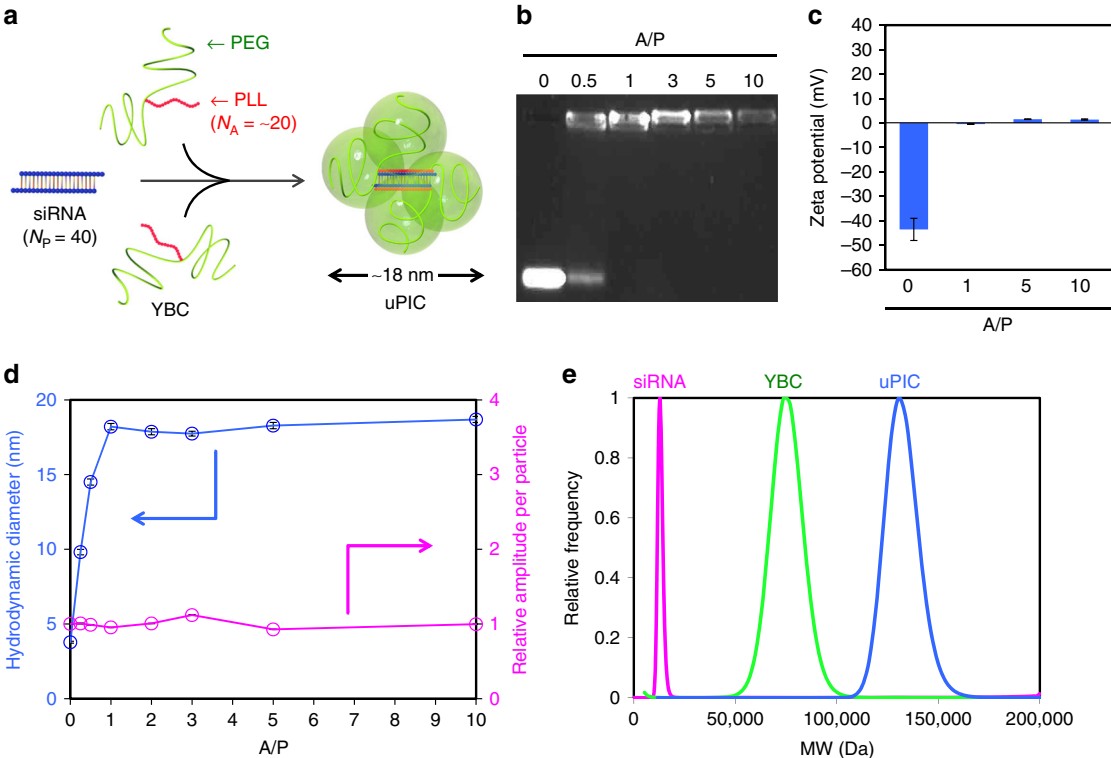

**Fig. 1** Construction and characterisation of uPIC from siRNA and YBC. **a** Schematic illustration of uPIC formation from the single siRNA and the dyad of YBCs through charge-matched ion-pairing. **b** Agarose gel retardation analysis of PIC samples prepared at varying A/P ratios. **c** Zeta potentials of PIC samples prepared at varying A/P ratios, determined by electrophoretic laser scattering. Data represent the means ± s.d. for three replicate measurements. **d** Hydrodynamic diameter of A647-siRNA complexed with YBC at varying A/P ratios, and the corresponding association number of siRNA per PIC, determined by FCS. Data represent the means ± s.d. for five replicate measurements. **e** Molecular weight histograms of free siRNA, free YBC, and uPIC prepared at an A/P ratio of 1, determined by a sedimentation velocity method in AUC

sample (iii) was directly prepared in the bloodstream by sequential intravenous injections of YBC (40 nmol) and A647-siRNA (2 nmol) and harvested in a manner similar to sample (ii). It should be noted that the strong fluorescence and specific absorbance of A647 dye permitted the determination of the $D_H$ and the MW of the siRNA/PICs, respectively, in the plasma. The $D_H$ of sample (iii), determined by FCS, was almost the same as the $D_H$ values of samples (i) and (ii) (Fig. 2h). Furthermore, the MW of the uPICs harvested from the bloodstream, determined by a sedimentation equilibrium method in AUC, was 156 kDa, which is comparable to that of the uPICs prepared in buffer. These results are consistent with the precise in situ formation of 2:1 uPICs without additional aggregation and micelle formation, even in the harsh environment of the bloodstream.

Next, we examined the longevity of siRNA after uPIC formation in the bloodstream. A647-siRNA/uPIC prepared at A/P = 10 was intravenously injected into mice, and the fluorescence intensity in the vein was continuously monitored by IVCLSM[14,15] (Fig. 3a–d and Supplementary Movies 3, 4). Based on the fluorescence intensity of A647, uPIC persisted in the circulation and had a blood half-life ($T_{1/2}$) of ~110 min, which was much longer than that of the naked siRNA ($T_{1/2}$ < 5 min) (Fig. 3e). The integrity of the circulating siRNA was also verified by IVCLSM and FRET, in which an siRNA labelled with A594 and A647 at the 3′- and 5′-ends of the antisense strand (FRET-siRNA), respectively, was used as an indicator of integrity. This method relies on the fact that intact FRET-siRNA produces an intense FRET signal attributable to the proximate A594 and A647, whereas the FRET signal decreases after siRNA degradation (Supplementary Fig. 5a). Prior to the in vivo experiment, we confirmed that the intensity of the FRET signal decreased

considerably after treatment of the FRET-siRNA with RNase A (Supplementary Fig. 5b). Intravenously injected FRET-siRNA/uPIC (A/P = 10) completely maintained the FRET signal for more than 120 min (Fig. 3f and Supplementary Movie 5), demonstrating the excellent stabilisation of siRNA in the bloodstream. The influence of free YBC on the blood circulation of uPIC was then examined by changing A/P from 1 to 10 (Supplementary Fig. 6a). The longevity of A647-siRNA in the bloodstream progressively increased with an increase in A/P, indicating that the excess of free YBC was essential for the longer blood circulation of siRNA. Importantly, the blood circulation profiles of A647-siRNA/uPICs at the higher A/P were apparently close to that of free A647-YBC (Supplementary Fig. 6b). These results are consistent with the dynamic equilibrium nature of uPIC; siRNA circulated in the bloodstream through the equilibrium of siRNA + YBC ⇄ uPIC, and the excess amount of free YBC shifted the equilibrium to the uPIC side for siRNA stabilisation. To further verify the effect of the advantage of the two-armed long PEG segment in the YBC in longevity of the uPIC, we compared the blood circulation properties of a series of uPICs prepared from block catiomers with different composition (Supplementary Fig. 7). Apparently, a uPIC prepared from a YBC comprising a PEG with two longer arms (MW = 2 × 37 kDa) enabled substantially longer blood circulation of A647-siRNA compared with a uPIC comprising a PEG with two shorter arms (MW = 2 × 21 kDa, $T_{1/2}$ = 56 min) or a uPIC with a linear PEG (MW = 73 kDa, $T_{1/2}$ = 41 min), presumably owing to the enhanced steric repulsive effect of the tetrahedrally configured two-armed PEG chains sufficiently covering PIC domain. We then further validated the impact of PIC formation between the siRNA and PLL segments by comparing uPICs with a covalent

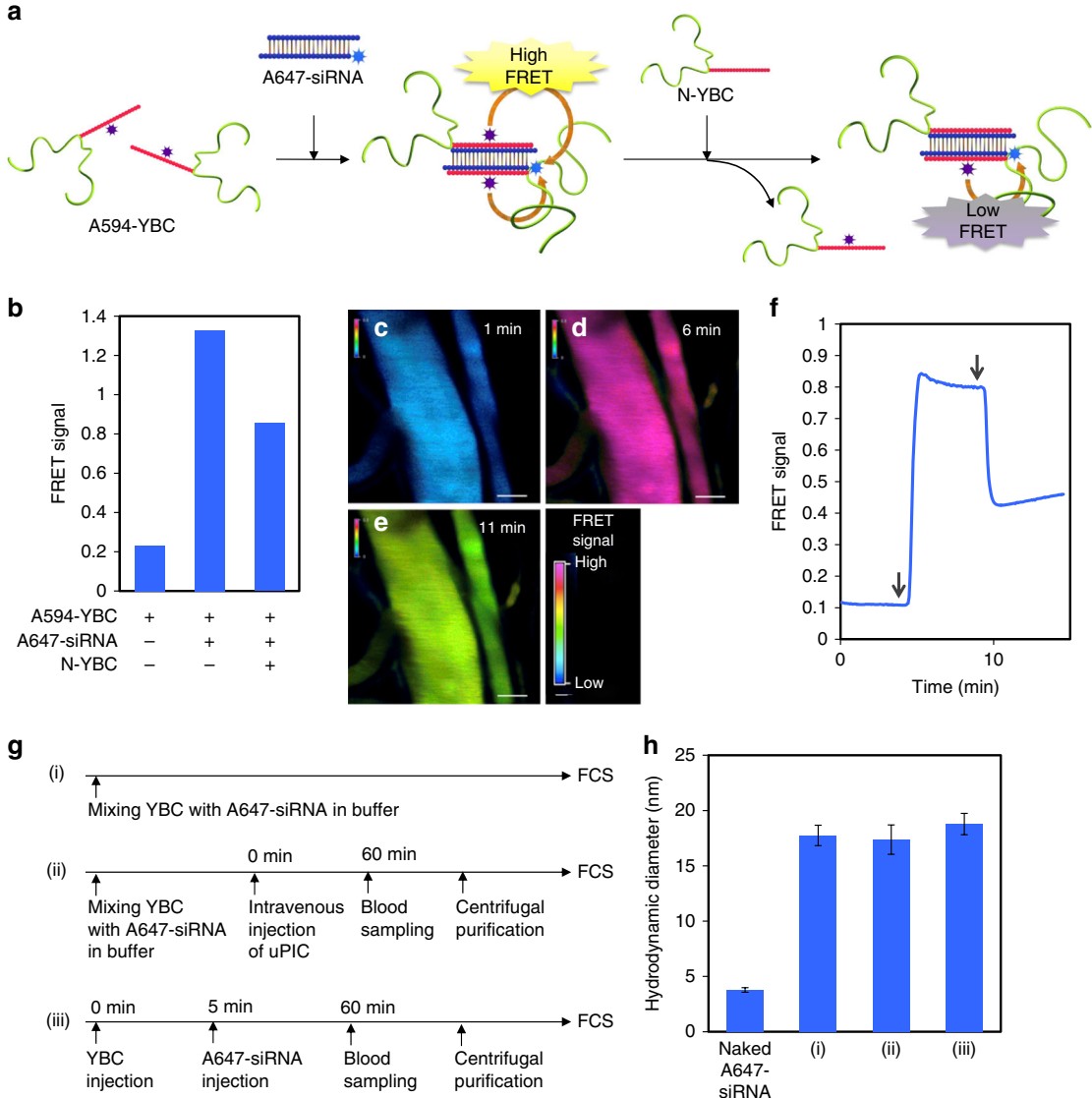

**Fig. 2** Dynamic and precise ion-pairing (rendezvous) of an siRNA with free YBCs. **a** Schematic illustration of the FRET-based analysis of dynamic ion-pairing of siRNA with free YBC (or the exchange reaction between uPIC-forming YBC and free YBC). **b** FRET analysis of uPIC formation and exchange reaction in buffer solution. **c–e** IVCLSM images of blood vessels in the mouse ear dermis treated with a stepwise systemic injection of A594-YBC (**c**), A647-siRNA (**d**), and N-YBC (**e**). The scale bars indicate 50 µm. **f** Time-dependent changes in the FRET signal obtained by quantitative analysis of the IVCLSM images acquired after A594-YBC injection. Left and right arrows indicate the injection of A647-siRNA and non-labelled YBC (N-YBC), respectively. **g** Sample preparation scheme for the comparative study of uPIC formation under varying conditions. **h** Hydrodynamic diameters of A647-siRNA complexed with YBC under varying conditions. Data represent the means ± s.d. for five replicate measurements

conjugate of siRNA comprising two-armed PEG chains (PEG-siRNA). The PEG-siRNA was obtained by a copper-free click conjugation between azide-functionalised two-armed PEG (MW = 2 × 37 kDa) and siRNA functionalised with dibenzylcyclooctyne at both 5′-ends after purification (Supplementary Fig. 8a, b). After treatment with RNase A for 30 min, almost all the siRNA in the PEG-siRNA was degraded, as observed for the unconjugated control siRNA (Supplementary Fig. 9a, b). In contrast, the uPIC exhibited excellent resistance to enzymatic degradation (Supplementary Fig. 9a, b), confirming the importance of PIC formulation for stability in biological fluids.

**Cancer-targetability of uPIC.** Many previous studies have demonstrated that nanoparticle carrier systems can efficiently accumulate in solid tumour models through the leaky tumour

vasculature and immature lymphatic drainage, termed enhanced permeability and retention (EPR) effect[16,17]. In this regard, recent studies on cancer pathophysiology have revealed that the nature of the tumour and its associated tissue significantly affects the accumulation and penetration profiles of nanoparticles extravasating from the bloodstream[4–9]. In particular, pancreatic cancer commonly involves a thick fibrotic stroma with hypovascularity, generating tumour microenvironments that are inaccessible to conventional nanoparticle carrier systems with a size of ~100 nm[5,6]. These characteristic tumour microenvironments are a current critical issue for cancer nanomedicine. In the present study, we selected a stroma-rich pancreatic cancer (BxPC3) model, which forms cancer cell nests surrounded by thick fibrosis (e.g. the fibrotic area: 40–50% in the tumour tissue[18]) and is therefore considered a good surrogate for modelling pancreatic desmoplasia in the clinic[5,19]. First, we compared the

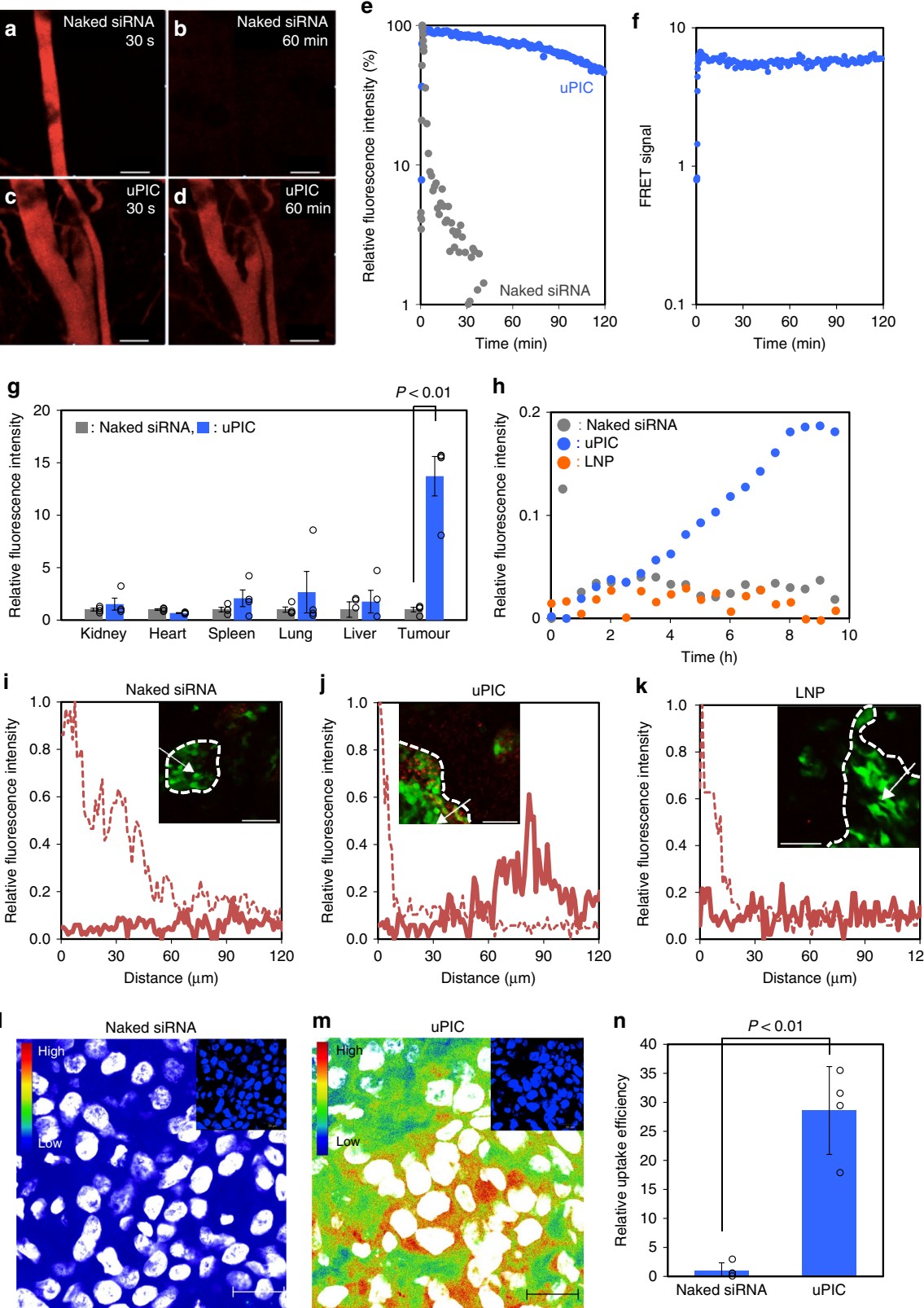

biodistribution of Cy5-labelled siRNA (Cy5-siRNA)/uPIC with naked Cy5-siRNA in subcutaneous tumour-bearing mice (Fig. 3g). The enhanced accumulation of uPIC in the tumours was obvious. There was ~15-fold more siRNA/uPIC than naked siRNA, demonstrating the high availability of uPIC in the stroma-rich tumour model. This high availability of uPIC is apparently

consistent with our previous results that small nanomedicine with a size of 30 nm can efficiently accumulate in the same tumour model, whereas counterparts above 50 nm did not show such accumulation[6]. Meanwhile, the distribution of uPICs in healthy organs did not differ significantly from that of naked siRNA. There were particularly low levels of accumulated uPICs in liver

**Fig. 3** Blood circulation and tumour accumulation profiles of uPIC. **a–d** IVCLSM images of circulating A647-siRNA in the blood vessels of the mouse ear dermis. Naked A647-siRNA at 30 s (**a**) and 60 min (**b**) after systemic injection. A647-siRNA/uPIC (A/P = 10) at 30 s (**c**) and 60 min (**d**) after systemic injection. The scale bars indicate 100 μm. **e** Time-dependent change in A647 fluorescence obtained by quantitatively analysing the IVCLSM images. **f** Time-dependent change in the FRET signal in the vein of the mouse ear dermis after systemic injection of FRET-siRNA/uPIC. **g** Biodistribution of naked Cy5-siRNA and Cy5-siRNA/uPIC 48 h after systemic administration. Fluorescence intensities were normalised to those obtained from naked siRNA. Data represent the means ± s.e.m. $n = 4$. **h** Time-dependent tumour accumulation profiles of naked Cy5-siRNA, uPIC, and Invivofectamine® LNP. The fluorescence intensity was obtained from the regions of interest (ROI) in the continuous IVCLSM images (Supplementary Fig. 11a–c). **i–k** Spatial profiling of fluorescence intensity derived from Cy5-siRNA in tumour tissues at the initial stage (dashed line) and 10 h (solid line) after systemic administration of naked siRNA (**i**), uPIC (**j**), and LNP (**k**). The fluorescence intensities were measured along the direction of the white arrow in the IVCLSM image inset (red: Cy5-siRNA; green: GFP-BxPC3 cells; scale bar: 100 μm) and normalised to the maximum value obtained at the initial stage. The cancer cell nest region was identified by GFP fluorescence derived from cancer cells, as depicted by the dashed line in the IVCLSM image inset. **l**, **m** Subcellular distribution of naked Cy5-siRNA (**l**) or Cy5-siRNA/uPIC (**m**). The circular voids are cellular nuclei identified from the nuclei-stained CLSM image inset. The scale bars indicate 20 μm. **n** Cellular uptake efficiency of siRNA in a spheroid culture of BxPC3 cells. The cellular uptake efficiency was determined by qRT-PCR as the amount of Argonaute 2-bound siRNA (or antisense strand) and normalised to the value obtained from the cells treated with naked siRNA. Data represent the means ± s.d. $n = 3$

and spleen (Supplementary Fig. 10), which was in sharp contrast to nanoparticle carrier systems, including lipid nanoparticles (LNPs), reported to date, which mainly accumulate in liver and spleen[20,21].

In-depth analyses of tumour accumulation were performed in a green fluorescent protein (GFP)-expressing BxPC3 (GFP-BxPC3) subcutaneous tumour model. Intravital images of GFP-positive tumour tissues within living animals were continuously recorded by IVCLSM after systemic administration of Cy5-siRNA in its naked form, uPICs, or Invivofectamine® LNP (Supplementary Fig. 11a–c and Supplementary Movies 6–8). Quantitative analysis revealed that the uPICs progressively accumulated in cancer cell nests over 10 h, whereas almost no accumulation was observed for naked siRNA and Invivofectamine® LNP (Fig. 3h). The Cy5-siRNA distribution in tumour tissues was quantitatively analysed to estimate the permeability of naked siRNA, uPICs, and LNP into the cancer cell nest (Fig. 3i–k). Whereas naked siRNA and LNP displayed no obvious fluorescence in the cancer cell nest, significant fluorescence was detected even at a depth of 80 μm in the cancer cell nest treated with uPICs, demonstrating the superior tissue permeability of uPICs. The tissue permeability of uPICs can presumably be ascribed to their small size and their PEG-derived ability to penetrate the stromal barrier of the pancreatic tumour tissue. It is worth noting that the MW of uPIC is comparable to that of human/murine immunoglobulin G (ca. 150 kDa[22,23]), which might favour both longevity in the bloodstream and tissue permeability. The subcellular distribution of uPICs in the tumour tissue was further determined by CLSM observations of frozen sections. Compared with the naked siRNA (Fig. 3l), a higher fluorescence intensity was detected surrounding the cell nuclei in the tumour tissue treated with uPICs (Fig. 3m), indicating that Cy5-siRNA/uPIC accumulated in the interior of the cancer cells after permeating the cancer cell nests. Therefore, we also examined the cellular uptake of uPICs using a spheroid culture of BxPC3 cells, which more closely resembled the in vivo 3D structure of solid tumour tissues than a conventional monolayer culture[24,25]. The cellular uptake efficiency of siRNA was determined by a two-step procedure: co-immunoprecipitation to extract the antisense strand bound to Argonaute 2 (an RNA-induced silencing complex) and quantitative reverse transcription polymerase chain reaction (qRT-PCR). These experiments were designed to obtain a better correlation between siRNA uptake and gene silencing efficiencies. The results revealed that the uptake efficiency of the uPIC was 30 times higher than that of naked siRNA (Fig. 3n).

The excellent penetration of uPICs into cancer cell nests encouraged us to investigate RNA interference (RNAi)-based cancer therapy in the stroma-rich subcutaneous tumour model.

*PLK1*, an essential kinase involved in the cell cycle, was chosen as a therapeutic target gene because its silencing can induce apoptosis in various cancer cells[26]. We first evaluated the gene silencing efficiency of human *PLK1*-targeted siRNA (sihPLK1)-loaded uPIC (sihPLK1/uPIC) in the tumour tissue. Systemically administered sihPLK1/uPIC elicited a significant reduction in the *PLK1* mRNA level compared with the control luciferase siRNA (siLuc)/uPIC (Fig. 4a) and in the *PLK1* protein level compared with naked siPLK1 and sihPLK1/Invivofectamine® LNP (Fig. 4b, c). These results demonstrated the sequence-specific gene silencing activity of siRNA/uPIC in the stroma-rich tumour. The sihPLK1-induced apoptotic effect in the tumour tissues was then analysed using a terminal deoxynucleotidyl transferase dUTP nick end labelling (TUNEL) assay (Fig. 4d–h). Significantly more TUNEL-positive cells were detected in the tumour tissue treated with sihPLK1/uPIC than in the tumour tissue treated with siLuc/uPIC or naked sihPLK1. The number of TUNEL-positive cells in the tumour tissue treated with siLuc/uPIC was similar to that in the tumour tissue treated with phosphate-buffered saline (PBS), suggesting the negligible toxic effect of siRNA/uPIC itself. The antitumour activity was evaluated by periodic and systemic injections of sihPLK1/ or siLuc/uPIC into subcutaneous BxPC3 tumour-bearing mice. Whereas siLuc/uPIC did not suppress tumour growth compared with the non-treated control, sihPLK1/uPIC significantly suppressed tumour growth (Fig. 4i) without any change in the body weight of the mice compared with the control (Fig. 4j). Thus, the antitumour activity of siRNA/uPIC via transporting apoptosis-inducing sihPLK1 was evidenced in the subcutaneous pancreatic tumour with rich stroma. Moreover, the comparable significant antitumour activity of siRNA/uPIC was also elicited by delivering a therapeutic siRNA encoding vascular endothelial growth factor (siVEGF) (Supplementary Fig. 12a), which can suppress tumour growth by disturbing angiogenesis in tumour tissues[27,28] without any change in body weight (Supplementary Fig. 12b). Then, the delivery capability of siRNA/uPIC was tested in a spontaneous pancreatic tumour model derived from a transgenic mouse with elastase 1-driven luciferase and SV40-derived large T antigens (oncomouse)[29] to demonstrate the RNAi and therapeutic efficacy of the present formulation to the non-xenograft model. siLuc/uPIC significantly reduced luciferase-derived luminescence intensity in a sequence-specific manner compared with a scrambled sequence siRNA (siScr)/uPIC (Fig. 4k and Supplementary Fig. 13). Ultimately, the oncomice treated with mouse *PLK1*-targeted siRNA (simPLK1)/uPIC had a significantly prolonged survival rate compared with those treated with siScr/uPIC (Fig. 4l), which was associated with a lower rate of liver metastases (Fig. 4m). It should be noted that the extension of the survival rate by simPLK1/uPIC, i.e. 86 and

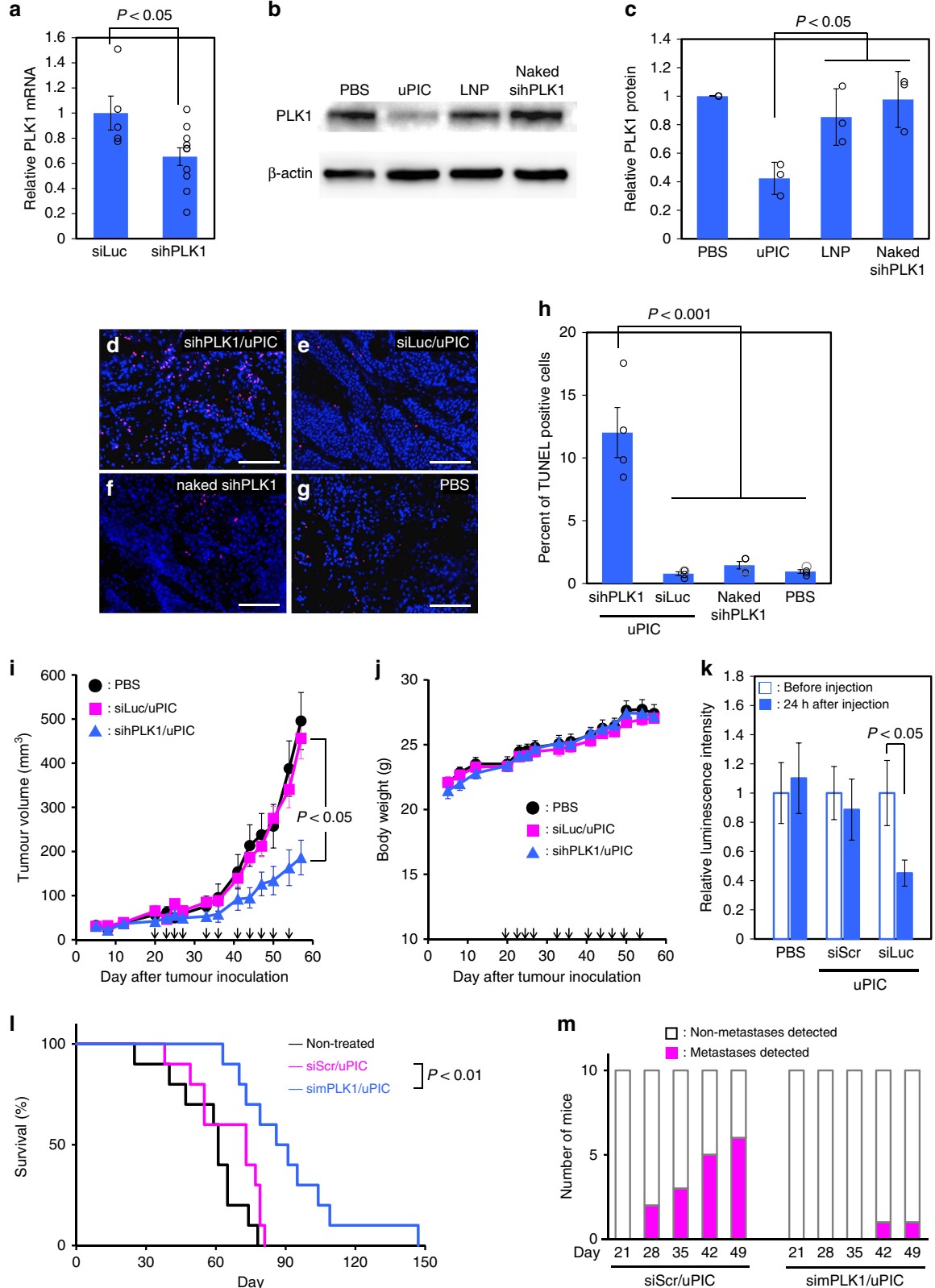

104 days for 50% and 20%, respectively, is apparently superior to that by oxaliplatin, e.g. 61 and 70 days at $2 \, mg \, kg^{-1}$ per week, previously reported[30]. These results demonstrate the strong potential of uPIC for systemic siRNA delivery to pancreatic tumour tissues. The blood laboratory data obtained for the mice treated with siScr/uPIC showed no signs of liver or kidney damage or of any immune or allergic response (Tables 1 and 2), indicating the potential safety of siRNA/uPIC.

We further extended our investigation of the delivery capability of uPIC to include systemic ASO (21-mer with 20 phosphorothioates) delivery. As with siRNA, charge-neutralised small PIC formation at $A/P \geq 1$ was confirmed by agarose gel

**Fig. 4** Gene silencing and therapeutic effects of uPIC. **a** *PLK1* mRNA level in tumour tissues harvested from subcutaneous BxPC3 tumour-bearing mice intravenously injected twice with uPIC at 24-h intervals. Data represent the means ± s.e.m. $n = 5$ for siLuc/uPIC and 10 for sihPLK1/uPIC. **b, c** *PLK1* protein level in the tumour tissues harvested from subcutaneous BxPC3 tumour-bearing mice intravenously injected with PBS, uPIC, Invivofectamine® LNP, and naked sihPLK1 thrice at 24-h intervals, as determined by western blotting (**b**), and the quantified results (**c**). Data represent the means ± s.d. $n = 3$.
**d**–**h** Apoptosis in subcutaneous BxPC3 tumour tissues from mice treated with sihPLK1/uPIC (**d**), siLuc/uPIC (**e**), naked sihPLK1 (**f**), or PBS (**g**) based on a TUNEL assay (red: apoptosis-positive cells; blue: cell nuclei stained with Hoechst 33342; scale bar: 100 μm), and the quantified results (**h**). Data represent the means ± s.e.m. $n = 4$. **i** Antitumour activity of uPIC intravenously administered multiple times (as shown by black arrows) to BxPC3 tumour-bearing mice. Data represent the means ± s.e.m. $n = 7$. **j** Body weight change in tumour-bearing mice treated with uPIC as described in (**i**). Data represent the means ± s.e.m. $n = 7$. **k** RNAi effect of uPIC in a spontaneous pancreatic tumour model. The luminescence intensity from luciferase-expressing pancreatic tumours was determined using an IVIS instrument 24 h after single systemic injection into oncomice and normalised to that obtained from the initial value before injection. Data represent the means ± s.e.m. $n = 9$–11. **l** Kaplan–Meier survival curves of non-treated oncomice and oncomice treated with simPLK1/ uPIC or siScr/uPIC. $n = 10$. **m** Number of oncomice with liver metastases at the designated days

**Table 1 Blood laboratory data for mice at 48 h after five systemic administrations (mean ± s.e.m. $n = 5$)**

|  | T.P. (g dL$^{-1}$) | Alb. (g dL$^{-1}$) | BUN (mg dL$^{-1}$) | Cr. (mg dL$^{-1}$) | Na (mEQ L$^{-1}$) | T.bil (mg dL$^{-1}$) | T.C. (mg dL$^{-1}$) | AST (IU L$^{-1}$) | ALT (IU L$^{-1}$) | LDH (IU L$^{-1}$) | ALP (IU L$^{-1}$) |
|---|---|---|---|---|---|---|---|---|---|---|---|
| Saline | 4.7 ± 0.1 | 3.0 ± 0.1 | 18 ± 0 | 0.09 ± 0.01 | 152 ± 2 | 0.05 ± 0.01 | 120 ± 3 | 52 ± 3 | 66 ± 9 | 229 ± 29 | 525 ± 13 |
| uPIC | 4.7 ± 0.1 | 2.8 ± 0.1 | 19 ± 1 | 0.10 ± 0.00 | 153 ± 2 | 0.04 ± 0.00 | 113 ± 2 | 50 ± 2 | 59 ± 5 | 227 ± 32 | 462 ± 10 |

*T.P.* total protein, *Alb* albumin, *BUN* blood urea nitrogen, *Cr.* creatinine, *T.bil* total bilirubin, *T.C.* total cholesterol, *AST* aspartate aminotransferase, *ALT* alanine aminotransferase, *LDH* lactate dehydrogenase, *ALP* alkaline phosphatase

**Table 2 Blood counts of mice at 48 h after five systemic administrations (mean ± s.e.m. $n = 5$)**

|  | WBC (g dL$^{-1}$) | Neut (%) | Lymph (%) | Mono (%) | Eo (%) | Baso (%) | Hb (g dL$^{-1}$) |
|---|---|---|---|---|---|---|---|
| Saline | 4800 ± 400 | 23 ± 1 | 71 ± 0 | 2.8 ± 0.4 | 2.9 ± 0.4 | 0.04 ± 0.04 | 16 ± 0 |
| uPIC | 5200 ± 600 | 19 ± 1 | 75 ± 1 | 3.3 ± 0.4 | 2.0 ± 0.4 | 0.00 ± 0.00 | 15 ± 0 |

*WBC* white blood cell, *Neut* neutrophil, *Lymph* lymphocyte, *Mono* monocyte, *Eo* eosinophil, *Baso* basophil, *Hb* haemoglobin

electrophoresis (Supplementary Fig. 14a) and FCS (Supplementary Fig. 14b). Importantly, the MWs of ASO/PIC at A/P = 1 and 20 were determined to be 83 ± 7 kDa and 84 ± 10 kDa (mean ± s. d., measured three times), respectively, using the AUC. These values correspond to the single pair comprising the ASO (MW = 7 kDa) and the YBC (MW = 76 kDa). Thus, the 1:1 uPIC arose owing to the charge neutralisation between the ASO ($N_P = 20$) and the YBC ($N_A = 20$) at their minimal association number. The dynamic equilibrium (or rendezvous) between uPIC and free YBC was also verified for the ASO/uPIC by similar FRET analyses using A594-YBC and A647-ASO in vitro (Supplementary Fig. 15a) and in vivo (Supplementary Fig. 15b). The longevity of A647-ASO in the bloodstream ($T_{1/2}$ = ~6 min for naked ASO) was dramatically enhanced after uPIC formation at A/P = 20 ($T_{1/2}$ = ~100 min) (Supplementary Fig. 16).

Brain tumours are widely known to be another intractable form of cancer that features the so-called blood–brain tumour barrier or blood–tumour barrier. Although the barrier characteristics, e.g. tight junction, narrow perivascular space, absence of fenestrae, and inactive pinocytosis, in the brain vasculature may be compromised in brain tumours[31], they still act as the obstacles restricting the extravasation of nanoparticles from the bloodstream to the tumour tissue[7–9]. In the present study, we challenged to treat an intracranial xenograft mouse model of glioma stem cells obtained from the surgical treatment of patients[32] by the use of the ASO/uPIC (A/P = 20). Systemic administration of A647-ASO/uPIC led to fairly efficient accumulation in the orthotopic murine xenografts, whereas almost no accumulation was detected for naked A647-ASO (Fig. 5a and Supplementary Fig. 17). Thus, uPIC successfully delivered ASO to the orthotopic brain tumour tissue, possibly because it was retained in the blood circulation for longer and was ultra-small, which increased its ability to penetrate the blood–brain tumour barrier. We then investigated ASO-based cancer therapy using the brain tumour model. In the present study, a long non-coding RNA, *TUG1*, was selected as a target molecule because *TUG1* silencing can suppress the self-renewal of glioma cells, potently repressing glioma cell growth[32]. Systemically administered *TUG1*-targeted ASO (asTUG1)/uPIC appreciably reduced the expression of *TUG1* (>90%) in the glioma tissue compared with a Luc-targeted ASO(asLuc)/uPIC control (Fig. 5b). Antitumour activity was evaluated by periodic and systemic injections of asTUG1/ or asLuc/uPIC into the glioma mouse model (Fig. 5c, d and Supplementary Fig. 18). The asTUG1/uPIC potently suppressed tumour growth compared with the asLuc/uPIC control, consistent with the *TUG1* silencing results. Moreover, the reduced expression patterns of stemness-associated genes (*MYC* and *SOX2*) and the reactivation of neuronal differentiation-associated genes (*BDNF*, *NGF*, and *NTF3*) were clearly observed in the asTUG1/uPIC-treated glioma tissues (Supplementary Fig. 19). Ultimately, the survival rate of the orthotopic brain tumour model mice treated with asTUG1/uPIC was dramatically extended (Fig. 5e). These results revealed that the uPIC successfully delivered therapeutic ASO in the patient-derived orthotopic brain tumour model.

## Discussion

A YBC was engineered to contain regulated positive charges (i.e. ~20) to selectively form a uPIC with a single molecule of siRNA or ASO. In this way, both SNAs were dynamically protected through in vivo rendezvous with YBCs, leading to stable

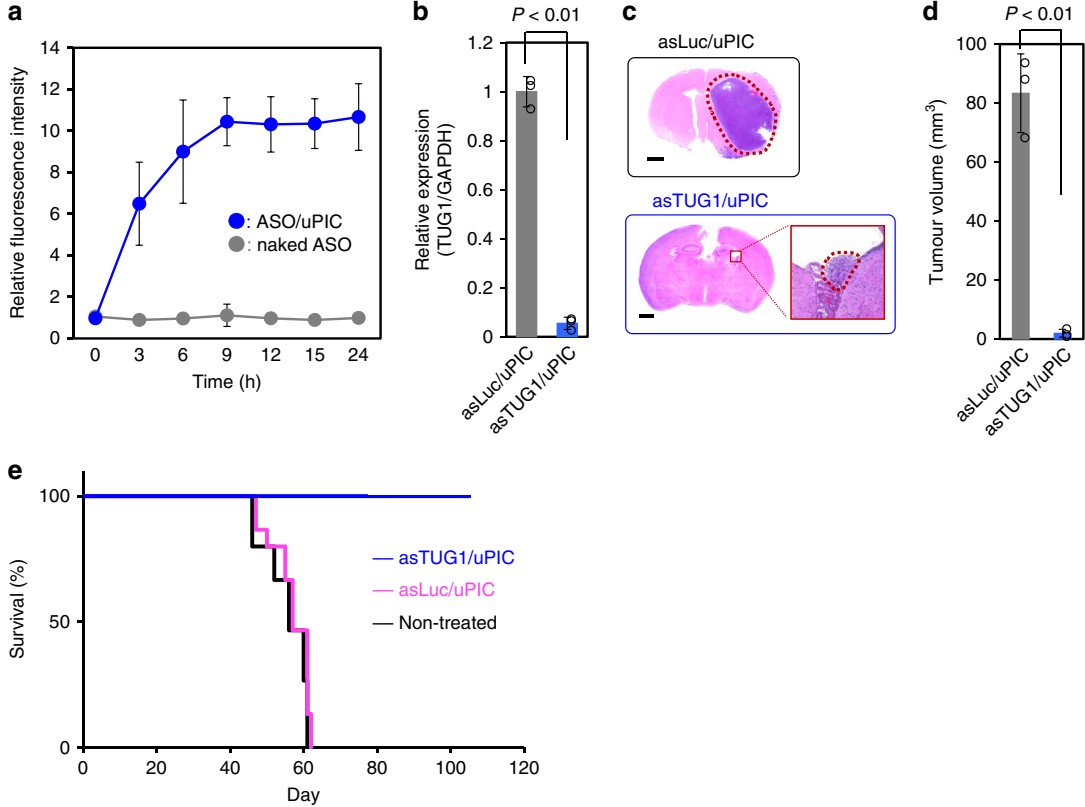

**Fig. 5** Delivery performances of ASO/uPIC. **a** Time-dependent accumulation profiles of naked A647-ASO and A647-ASO/uPIC (A/P = 20) in a patient-derived orthotopic brain tumour model, as measured by IVIS. Data represent the means ± s.d. $n = 3$. **b** *TUG1* expression level in brain tumour tissue. Data represent the means ± s.d. $n = 3$. **c** Representative HE-stained whole brain sections at 4 weeks after treatment by intravenous injection of asLuc/ or asTUG1/uPIC. The tumour areas are surrounded by the red dotted line. Scale bars indicate 1 mm. **d** Tumour volumes at 4 weeks after treatment by intravenous injection of asLuc/ or asTUG1/uPIC. Data represent the means ± s.d. $n = 3$. **e** Kaplan–Meier survival curves of orthotopic brain tumour model mice treated with asTUG1/uPIC or asLuc/uPIC. uPIC samples (25 µg ASO per mouse) were intravenously administrated every three days (total 10 doses). $n = 10$

circulation in the bloodstream. Subsequently, the uPICs, which were as small as ~18 nm, penetrated deep into the stroma-rich pancreatic tumour tissues without accumulating in the liver or spleen and exerted significant antitumour activity through the selective silencing of an apoptosis-related gene with negligible adverse side effects. Furthermore, the uPIC efficiently delivered ASO in an orthotopic brain tumour model, which may have featured a blood–brain tumour barrier, and exhibited excellent antitumour activity in the model by delivering asTUG1. The uPIC formulation was simple and was readily achieved by mixing aqueous solutions of the YBC and SNA. This facile preparation procedure is indeed a great advantage from the perspective of pharmaceutical development. Although the detailed mechanisms involved in the cellular internalisation and the subsequent intracellular processing of uPIC remain to be clarified, its unique propensities overcome current issues associated with oligonucleotide therapeutics, including blood circulation stability and the ability to penetrate tumours with barrier characteristics. While the present SNA delivery strategy of uPIC needs to be further verified for a wide variety of tumour models to gain better insight into the relevance to clinical human cancers, it demonstrates the strong potential for the systemic use of oligonucleotide therapeutics against intractable cancers with tight access barriers.

## Methods

**Materials**. Two-armed, α-methoxy-ω-amino poly(ethylene glycol) (two-armed PEG-NH₂, $M_n = 2 \times 37$ kDa) was obtained from NOF (Tokyo, Japan). ε-Tri-fluoroacetyl-L-lysine N-carboxy-anhydride (L-Lys(TFA)-NCA) was prepared by the

Fuchs-Farthing method using triphosgene[33,34]. Standard PEG molecules ($M_n$ = 0.96, 1.47, 4.25, 7.1, 12.6, 23, 50, and 101 kDa) were used for calibration in the HPLC measurements and were purchased from Polymer Laboratories, Ltd. (Church Stretton, UK) and Tosoh Corp. (Tokyo, Japan). Sterile HEPES (1 M, pH 7.3) was purchased from Amresco (Solon, OH, USA). UltraPure™ Agarose, Alexa Fluor® 594 NHS Ester, and Invivofectamine® Reagent were purchased from Life Technologies (Carlsbad, CA, USA). PD-10 desalting columns and Cy5 dye were purchased from GE Healthcare (Little Chalfont, UK). A pancreatic adenocarcinoma cell line, BxPC3, was obtained from the American Type Culture Collection (Manassas, VA, USA) and used after confirming the absence of mycoplasma contamination. This cell line is not listed in the database of commonly mis-identified cell lines maintained by ICLAC (ver. 7.2), and information about it is available at NCBI Biosample. GFP-expressing BxPC3 cells, authenticated by short tandem repeat-PCR (STR-PCR), were purchased from AntiCancer Japan Inc. (Chiba, Japan). Dorsal skin fold chambers were purchased from APJ Trading Co. Inc. (Ventura, CA, USA). Tissue-Tek® O.C.T.™ Compound was purchased from Sakura Finetek (Tokyo, Japan). Non-labelled siRNA and Cy5-siRNA were synthesised by Hokkaido System Science Co., Ltd. (Hokkaido, Japan). A647-siRNA and A594/A647 dual-labelled siRNA (FRET-siRNA) were synthesised by Gene Design Inc. (Osaka, Japan). All ASOs were synthesised by Gene Design Inc. The structures and sequences[28,32,35–37] of siRNAs and ASOs used in the present study are summarised in Supplementary Table 1. In Cy5- and A647-siRNAs, the dyes were attached to the 5′ end of the sense strand of siLuc. In FRET-siRNA, the A594 and A647 dyes were attached to the 3′ and 5′ ends of the antisense strand of siLuc, respectively. In A647-ASO, the A647 dye was attached to the 3′-end of asLuc. It should be noted that BLAST® was used to confirm that the sequence of siScr did not fully match the murine mRNAs. All the primers used in this study are listed in Supplementary Table 2.

**Animal experiments**. Animal experiments using healthy mice and subcutaneous tumour-bearing mice were performed in accordance with the guidelines for animal experiments at The University of Tokyo, Japan. Experiments involving oncomice were performed in accordance with the ethics committee of the Innovation Center of NanoMedicine, Japan, and experiments involving orthotopic brain

tumour-bearing mice were performed in accordance with the protocols approved by the Institutional Animal Care and Use Committee of Nagoya University Graduate School of Medicine, Japan, respectively. All experiments received ethical approval from the corresponding institute committee. BALB/c mice and BALB/c nude mice (mainly 6-week-old males) were purchased from Charles River Laboratories Japan (Yokohama, Japan) and used for sample injection or tumour inoculation within 1 week. All mice were treated with isotonic sample solutions and subjected to isoflurane-inhaled anaesthesia during the experiments. The number of mice used in each experiment was chosen by assuming the minimum to obtain statistical significance. Tumour-bearing mice were randomly allocated to each group to fulfil the variation among mean tumour sizes in each group within $2 \times$ s.d. Blinding was not used in any of the experiments.

**Synthesis of Y-shaped block catiomer (YBC).** A block copolymer—two-armed PEG-*block*-poly(L-lysine) (two-armed PEG-PLL or YBC)—was synthesised via the ring-opening polymerisation of L-Lys(TFA)-NCA initiated by the terminal primary amino group of two-armed PEG-NH$_2$ and subsequent deprotection of TFA protective groups[11,34]. Briefly, two-armed PEG-NH$_2$ (0.74 g, 10 μmol) and L-Lys (TFA)-NCA (70 mg, 260 μmol) were separately dissolved in distilled DMF (7.2 mL and 5.0 mL, respectively) containing 1 M thiourea. The L-Lys(TFA)-NCA solution was added to the PEG solution and the mixture was stirred at 25 °C for 1 day. The reaction mixture was poured dropwise to diethyl ether (200 mL) and reprecipitated twice with methanol/diethyl ether. The precipitate was collected by filtration to yield two-armed PEG-PLL(TFA) (0.50 g). The two-armed PEG-PLL(TFA) (100 mg) was dissolved in methanol (10 mL), mixed with 1 M NaOH aq. (1 mL), and stirred overnight at 35 °C. The reaction solution was dialyzed against 10 mM HCl aq. for 8 h and then distilled water for 24 h. The lyophilization of dialyzed solution afforded 80 mg of two-armed PEG-PLL. The synthesised two-armed PEG-PLL was characterised using size exclusion chromatography (SEC) (Supplementary Fig. 1a) and $^1$H NMR (Supplementary Fig. 1b). The SEC measurements were performed using a JASCO HPLC system equipped with a UV detector (UV-1575, JASCO, Tokyo, Japan) and a Superdex 200 10/300 GL column (GE Healthcare) eluted with 10 mM acetate buffer (pH 3.3) containing 0.5 M sodium chloride (flow rate: 0.5 mL min$^{-1}$, room temperature). The MW distribution ($M_w/M_n$) of the two-armed PEG-PLL was 1.07 based on standard PEG molecules and calculated using JASCO HPLC system software. The $^1$H NMR spectrum was recorded using a JEOL EX300 spectrometer (JEOL, Tokyo, Japan). The degree of polymerisation of the PLL segment (DP$_{PLL}$) was calculated to be ~20 based on the peak intensity ratios of the oxyethylene protons of PEG ($-OCH_2CH_2-$, $\delta = \sim 3.5$ ppm) to the β, γ, and δ methylene protons of the lysine side chain units ($-CH_2CH_2CH_2-$, $\delta = 1.3–1.9$ ppm) in the $^1$H NMR spectrum (D$_2$O, 20 °C). The A594-labelled two-armed PEG-PLL (A594-YBC) was synthesised using Alexa Fluor 594 NHS ester. The two-armed PEG-PLL (20 mg) was dissolved in 0.1 M sodium bicarbonate buffer (pH 8.3, 2.0 mL) and added to Alexa Fluor 594 NHS ester dissolved in DMF (10 mg mL$^{-1}$, 22 μL) with stirring. After a 2-h incubation at room temperature with stirring, the reaction solution was purified using a PD-10 desalting column equilibrated with 1 M NaCl. The two-armed PEG-PLL-containing eluent was dialysed against pure water for 2 days, followed by lyophilisation to obtain A594-YBC.

**Preparation of uPIC from siRNA and YBC.** The siRNA/uPICs were generally prepared as follows. siRNA and YBC were separately dissolved in 10 mM HEPES buffer (pH 7.4) at concentrations of 60 μM and 329 μM, respectively. These solutions were simply mixed by pipetting at the designated mixing ratios (or A/P ratios; final siRNA concentration: 10 μM). The mixture (or PIC solution) was stored for 30 min at ambient temperature before use.

**Gel retardation analysis of PIC.** uPICs were prepared from siLuc at A/P ratios of 0, 0.5, 1, 3, 5, and 10 and diluted 100 times with 10 mM HEPES buffer. Glycerol was added to the uPIC solutions (final glycerol concentration: 5 vol%; final siRNA concentration: 50 nM). The uPIC samples were electrophoresed on an agarose gel (1 wt% agarose gel; 1 × TAE buffer; 140 V, 30 min). siRNA on the gel was stained with ethidium bromide and visualised using a FAS-III imager (Toyobo Life Science, Osaka, Japan).

**Zeta potential measurement.** The zeta potentials of the PIC samples prepared from siLuc were measured using a Zetasizer Nano-ZS instrument (Malvern Instruments, Malvern, UK) equipped with a He-Ne laser (532 nm). The naked siRNA samples (A/P = 0) were prepared with 10 μM siRNA in 10 mM HEPES buffer (pH 7.4). The PIC samples (A/P = 1, 5, and 10) were prepared with 20 μM YBC in 10 mM HEPES buffer (pH 7.4). Sample solutions (700 μL) were added to a folded capillary cell (Malvern Instruments). The obtained electrophoretic mobility was converted to the zeta potential using the equipment software. The results are represented as the mean and s.d. obtained from three measurements.

**Fluorescence correlation spectroscopy (FCS).** FCS analysis was carried out using a combination system consisting of a ConfoCor3 module and LSM510 equipped with a C-Apochromat 40×, N.A. 1.2 water immersion objective (Carl Zeiss, Oberkochen, Germany). A He-Ne laser (wavelength: 633 nm) was used for excitation of the A647 dye in A647-siRNA. The PICs were prepared from YBC and

A647-siRNA at various A/P ratios, as described above, and were then diluted to 100 nM siRNA with 10 mM HEPES buffer (pH 7.4) containing 150 mM NaCl. Thereafter, each sample was placed in an 8-well Laboratory-Tek chamber (Nalgene Nunc International, Rochester, NY, USA). The FCS measurements were performed with a sampling time of 15 s and a repeat time of 5, and the obtained autocorrelation curves of the samples were converted to the diffusion time using ConfoCor3 software. The diffusion time was further converted to the diffusion coefficient ($D_C$) using Cy5 dye as a reference. The $D_H$ was calculated using the Stokes-Einstein equation:

$$D_H = k_B T / 3\pi\eta D_C \qquad (1)$$

where $k_B$ is the Boltzmann constant, $T$ is the temperature, and $\eta$ is the solvent viscosity. The association number of siRNA per uPIC (AN$_{siRNA}$) was calculated using the following equation, where $N$ is the fluorescence amplitude per particle observed in a confocal volume:

$$AN_{siRNA} = N_{nakedsiRNA}/N_{PIC} \qquad (2)$$

For size measurement of PIC harvested from the bloodstream, two samples were prepared as illustrated in Fig. 2g. In sample (ii), a uPIC sample was prepared at A/P = 10 (2 nmol and 40 nmol YBC) in 10 mM HEPES buffer (pH 7.4) and intravenously administered at $t = 0$. In sample (iii), YBC (40 nmol) was intravenously administered at $t = 0$, followed 5 min later by an additional injection of A647-siRNA (2 nmol). In samples (ii) and (iii), the blood sample (500 μL) was harvested from the postcaval vein of the anesthetised mice at $t = 60$ min and centrifuged at $800 \times g$ for 10 min at 25 °C to obtain the plasma. The plasma was diluted 10 times with PBS to adjust the fluorescence intensity to the detection range appropriate for FCS analysis. The $D_H$ of the uPIC harvested from the bloodstream was determined as described above.

**Densimetry.** The densities of the siRNA (siLuc) and YBC solutions were measured using a density meter (DMA4500/DMA5000, Anton Paar, Graz, Austria). Both solutions were diluted to 1 and 5 mg mL$^{-1}$ with 10 mM HEPES buffer (pH 7.4) containing 150 mM NaCl, and the density measurements were performed at 20 °C. The partial specific volume of component $i$ (PSV$_i$) was calculated using equation 3.

$$PSV_i = (1 - d\rho/dc)/\rho_0, \qquad (3)$$

where $\rho_0$ is the buffer density, $\rho$ is the solution density, and $c$ is the solute concentration. PSV$_{PIC}$ was determined as the mass average of PSV$_{siRNA}$ and PSV$_{YBC}$:

$$PSV_{PIC} = (M_{YBC} \times PSV_{YBC} + M_{siRNA} \times PSV_{siRNA})/(M_{YBC} + M_{siRNA}) \qquad (4)$$

where $M_{YBC}$ and $M_{siRNA}$ are the masses of YBC and siRNA, respectively, in the solution.

**Analytical ultracentrifuge (AUC) measurement.** A sedimentation velocity method in AUC (SV-AUC) and a sedimentation equilibrium method in an AUC (SE-AUC) were performed with a Beckman Optima XL-A analytical ultracentrifuge equipped with absorbance optics (Beckman Coulter, CA, USA). In SV-AUC, the concentrations of naked siRNA (siLuc) and YBC solutions were adjusted to 3 μM and 13 μM, respectively, using 10 mM HEPES buffer (pH 7.4) containing 150 mM NaCl. The concentration of the PIC solution prepared from siRNA and YBC at A/P = 1 was adjusted to 3 μM siRNA. The absorbance at 260 nm (for siRNA and uPIC) and 230 nm (for YBC) was repeatedly measured every 10 min as a function of the centrifugal radius ($r$) in a 142,000 × g centrifugal field at 20 °C for 10 h. Acquired data sets were analysed using SEDFIT software to determine the MW distribution[38]. In SE-AUC, the concentrations of uPICs prepared at A/P = 1, 5, and 10 were adjusted to 0.6 μM siRNA with 10 mM HEPES buffer (pH 7.4) containing 150 mM NaCl. The absorbance at 260 nm was measured as a function of $r$ at 20 °C after 72 h of centrifugation. The measurements for each uPIC were analysed using ORIGIN software supplied by Beckman to determine the MW by fitting to the following equation based on the values of PSV$_{PIC}$ and the buffer density:

$$\ln(C(r)/C(r_0)) = MW \times (1 - PSV_{PIC}) \times \rho_0 \times \omega^2 \times (r^2 - r_0^2)/2RT \qquad (5)$$

where $C(r)$ is the concentration of siRNA at $r$, $\omega$ is the rotational speed, $R$ is the gas constant, $T$ is the temperature, and $r_0$ is the minimum centrifugal radius for absorbance measurements.

For determination of the MW of uPIC harvested from the bloodstream, YBC (180 nmol) was intravenously administered to the mice, followed 5 min later by an additional intravenous injection of A647-siRNA (9 nmol). At 5 min after the second injection, the blood samples were harvested from the postcaval vein. The plasma was then collected as described above and diluted five times with PBS, followed by employing SE-AUC with the recording of absorbance at 650 nm as a function of r at 4 °C after centrifugation for 48 h. The MW of the uPIC was determined as described above, by assuming that the A/P of the uPIC was maintained at the initial value.

**Fluorescence resonance energy transfer (FRET) analyses.** The exchange reaction between uPIC-forming YBC and free YBC in a buffer solution was monitored by FRET analysis using an FP-6600 spectrofluorometer (JASCO).

Fluorescence intensities at 622 and 669 nm with excitation at 561 nm were measured for the following: the A594-YBC solution (2 μM); the mixture of A594-YBC and A647-siRNA (2 μM and 1 μM, respectively); and the mixture of A594-YBC, A647-siRNA, and N-YBC (2 μM, 1 μM, and 2 μM, respectively), which was prepared by adding N-YBC to the mixture of A594-YBC and A647-siRNA. The FRET signal was defined as the fluorescence intensity ratio: [intensity at 669 nm]/[intensity at 622 nm].

The exchange reaction in the bloodstream was monitored using a similar FRET method with a Nikon A1R confocal laser scanning microscope system attached to an upright ECLIPSE FN1 equipped with a CFI Plan Apo λ 20 × objective (Nikon Corp., Tokyo, Japan) (Supplementary Movie 2). Each BALB/c mouse was anesthetised with isoflurane and placed onto a temperature-controlled pad (Thermoplate; Tokai Hit Co., Ltd., Shizuoka, Japan) that was integrated into the microscope stage. The mice were subjected to a lateral tail vein catheterisation with a 30-gauge needle (Dentronics Co., Ltd., Tokyo, Japan) connected to a nontoxic, medical grade polyethylene tube (Natsume Seisakusho Co., Ltd., Tokyo, Japan). The earlobe dermis was observed following fixation beneath a coverslip with a single drop of immersion oil. Solutions of A594-YBC (200 μM, 100 μL), A647-siRNA (40 μM, 50 μL), and N-YBC (200 μM, 100 μL) were sequentially administered through a preset catheter into the mice at $t = 0$, 4, and 9 min, respectively. Fluorescent signals excited at 561 nm were recorded as spectral images in the 583–703-nm range with a spectral resolution of 6 nm. For quantification, the spectral images were linearly unmixed to the A647 and A594 spectra. Regions of interest (ROIs) were manually defined in the region of the vein in the spectral image, and the mean FRET signal, defined as [area under the curve of the A647 spectrum]/[area under the curve of the A594 spectrum], was calculated against time. The image processing was conducted using NIS Elements software (Nikon Corp.).

siRNA integrity was also monitored by FRET analysis using A594/A647 dual-labelled 'FRET-siRNA' and an IVCLSM system similar to the FRET analysis system used for the polymer exchange reaction. Naked FRET-siRNA or FRET-siRNA/uPIC prepared at A/P = 10 was intravenously injected at 24 μg FRET-siRNA/mouse. FRET-siRNA fluorescence signals were detected using a 561-nm excitation laser and lambda stacks in the 580–706-nm range at a bandwidth of 6 nm. The spectral images obtained were linearly unmixed to the A647 and A594 spectra. The FRET signal was defined as [area under the curve of the A647 spectrum]/[area under the curve of the A594 spectrum] and used as an indicator of siRNA integrity.

**Blood circulation test**. The circulation of siRNA in the bloodstream was investigated using A647-siRNA and an IVCLSM system equipped with a 20 × objective, a 640-nm diode laser, and a band-pass emission filter of 700/75 nm. The pinhole diameter was set to produce a 10-μm optical slice. As mentioned above, each BALB/c mouse was placed on the microscope stage, and the earlobe dermis was observed. Naked A647-siRNA or A647-siRNA/uPIC prepared at A/P = 10 was intravenously injected at 24 μg A647-siRNA per mouse. Fluorescence images were recorded in video mode for 3 min (30 frames s⁻¹), followed by snapshots every 1 min thereafter. The relative fluorescence intensity (%) was calculated for the ROIs set within the vein using equation 6.

$$\text{Relative fluorescence intensity} = 100 \times [[(\text{intensity at the indicated time-point}) - (\text{background intensity})]/[(\text{maximum intensity}) - (\text{background intensity})]] \tag{6}$$

**Biodistribution of siRNA in subcutaneous tumour-bearing mice**. A human pancreatic adenocarcinoma cell line, BxPC3, was subcutaneously injected at a concentration of $1 \times 10^8$ cells per mouse into the right rear flank of each BALB/c nude mouse. The tumours were allowed to grow to similar volumes (~50 mm³) for 3 weeks before sample injection. Naked Cy5-siRNA or Cy5-siRNA/uPIC prepared at A/P = 10 was intravenously injected into the tail vein of each mouse at 72 μg Cy5-siRNA per mouse. After 48 h, the tumours and organs were excised and quantitatively analysed with an IVIS instrument (PerkinElmer, Waltham, MA, USA). The obtained fluorescence intensity was first normalised to the area of each tissue or organ (mm²) (Supplementary Fig. 10). The relative fluorescence intensity was then calculated by normalisation to that obtained from naked Cy5-siRNA (Fig. 3g).

**siRNA distribution in subcutaneous pancreatic tumour tissue**. GFP-BxPC3 cells were subcutaneously injected at a concentration of $1 \times 10^8$ cells per mouse into the dorsal skin of each BALB/c nude mouse. Once the tumours had reached ~50 mm³ in volume, they were excised and cut into 1-mm dice. The tumour dice were used to prepare a dorsal skin fold chamber (DSFC) model of the BALB/c nude mice. After vascularisation was observed in the DSFC, the mice were placed on the microscope stage, and the DSFC was observed using an IVCLSM system equipped with a 20 × objective. Naked Cy5-siRNA, Cy5-siRNA/uPIC prepared at A/P = 10, or Cy5-siRNA/Invivofectamine® LNP prepared according to the manufacturer's protocol, was intravenously injected at 24 μg Cy5-siRNA/mouse. GFP and Cy5 fluorescent signals were detected using 488 and 640 nm excitation lasers and band-pass emission filters of 525/50 and 700/75 nm, respectively. Fluorescence images were acquired by obtaining snapshots every 30 min (Supplementary Fig. 11a–c).

The relative fluorescence intensity was calculated as described for the blood circulation test.

**Subcellular distribution of siRNA in subcutaneous tumour**. BxPC3 cells were subcutaneously injected at a concentration of $1 \times 10^8$ cells per mouse into the right rear flank of each BALB/c nude mouse. The tumours were allowed to grow for 8 weeks before intravenous injection of naked Cy5-siRNA or Cy5-siRNA/uPIC (A/P = 10) into the tail vein of each tumour-bearing mouse (72 μg siRNA per mouse). The tumours were dissected 24 h after administration, fixed with 4% buffered formalin, and embedded in O.C.T. compound after decalcification with 10% EDTA. The frozen frontal sections were cut at −20 °C using a cryostat (CM3050s, Leica, Wetzlar, Germany). The sections were dried at room temperature, washed with PBS, and stained with Hoechst 33342 (Life Technologies) for 15 min. The distribution of Cy5-siRNA in the tumour tissues was observed using a CLSM (LSM780, Carl Zeiss).

**Cellular uptake study using a spheroid culture**. BxPC3 cells (20,000 cells per well) were seeded into a 96-well spheroid culture plate, the Cell-able® Oncology 96-well plate (BP-96-R800, Toyo Gosei Co., Ltd, Tokyo, Japan), in growth medium and incubated for 5 days. Naked siRNA (siLuc) or uPIC (A/P = 10) was then added to each well at a concentration of 100 nM siRNA and incubated for 24 h. The cells were lysed with lysis buffer (50 mM Tris, 200 mM NaCl, 0.5% Triton X-100, 2 mM EDTA, pH 7.5) containing a protease inhibitor cocktail (P8340, Sigma–Aldrich, St. Louis, MO, USA)[39]. The lysate was centrifuged at 13,000 × g and 4 °C for 20 min, and the supernatant was collected. In parallel, anti-Argonaute 2 antibody (1 μg, 015-22031, Wako Pure Chemical Industries, Ltd., Osaka, Japan) was incubated overnight with 90 μL of Dynabeads® Protein G for immunoprecipitation (1003D, Thermo Fisher Scientific Inc., Waltham, MA, USA) in 200 μL of PBS at 4 °C. The antibody-incorporated beads were washed twice with lysis buffer and incubated overnight with 1300 μg of the protein supernatant at 4 °C. The beads were washed three times with lysis buffer and then incubated with 0.1 M glycine buffer (30 μL, pH 2.3) for elution. The eluent was mixed with 1 M Tris buffer (30 μL, pH 8.0) containing 20 U mL⁻¹ of proteinase K (EO0491, Thermo Fisher Scientific Inc.) for pH adjustment, incubated at 60 °C for 20 min to digest the proteins, and further incubated at 95 °C for 10 min for inactivation. The antisense strands of siRNA in the sample solution were reverse-transcribed using a Synthetic siRNA Quantitation Core Kit (Takara Bio Inc., Shiga, Japan) and quantified by FastStart Universal SYBR Green Master (Roche, Basel, Switzerland) according to the manufacturers' protocols.

**qRT-PCR**. BxPC3 cells were subcutaneously injected at a concentration of $1 \times 10^8$ cells per mouse into the right rear flank of each BALB/c nude mouse. The tumours were allowed to grow for 8 weeks prior to sample administration. siLuc/uPIC or sihPLK1/uPIC (A/P = 10) was intravenously injected twice into the tail vein of each tumour-bearing mouse at 24-h intervals (24 μg siRNA per mouse per shot). Thereafter, each tumour was excised and homogenised using Multi-Beads Shocker (Yasui-Kikai, Osaka, Japan). The RNAs were purified from the homogenised tissues using Isogen (Nippon Gene, Tokyo, Japan), chloroform, isopropanol, and 70% ethanol. The purified RNAs were converted to cDNA using ReverTra qPCR RT Master Mix (Toyobo Life Science) and amplified by qRT-PCR using the Taqman Fast Universal PCR Master Mix and the 7500 Fast Real-Time PCR System (Life Technologies). The primers comprised the *hPLK1* primer (Hs00153444_m1) and an *18 S* control primer (Hs99999901_s1) (Life Technologies).

**Western blot analysis**. Subcutaneous BxPC3 tumour tissues prepared as described above were excised 24 h after thrice-daily intravenous administration of the samples to the tumour-bearing mice (24 μg siRNA per mouse per shot). The excised tumour tissues (5 mg) were mixed with 1 mL of RIPA buffer (Sigma–Aldrich) containing a proteinase inhibitor cocktail (Sigma–Aldrich) and then homogenised by sonication. The lysates were agitated at 4 °C for 2 h and centrifuged at 13,000 × g and 4 °C for 20 min. The proteins in the supernatant were separated by SDS-PAGE (Bolt 8% Bis-Tris Plus, Invitrogen, Carlsbad, CA, USA) and transferred to a polyvinylidene difluoride membrane. The membrane was exposed to human *PLK1* antibody (#4513, Cell Signaling Technology, Danvers, MA, USA) or human *β-actin* antibody (#4967, Cell Signaling Technology) for 1 h, washed with Tris-buffered saline containing 0.1% Tween 20, and incubated with HRP-linked secondary antibody (#7074, Cell Signaling Technology) for 1 h at a dilution of 1:1000 for all antibodies. The protein bands were visualised using ECL Prime Western Blotting Detection reagent (Amersham Little Chalfont, UK) and a chemiluminescent imager (LAS-4000, Fujifilm, Tokyo, Japan). The band intensity was determined using Multi Gauge software (Fujifilm). The experiments were repeated for three different tumours treated with each sample (n = 3). The representative image was shown in Fig. 4b and its uncropped scans including MW markers are shown in Supplementary Figure 20.

**Apoptosis detection (TUNEL) assay**. The tumour tissues excised from the mice treated with thrice-daily intravenous injections of the samples, as described in the western blot analysis section, were perfused with 4% paraformaldehyde (PFA), fixed overnight in 4% PFA, and cryoprotected using consecutive solutions of 10, 15,

and 20% sucrose per PBS. After freezing in O.C.T. compound, 10-μm-thick sections were prepared. The sections were stained with an anti-GFP rabbit monoclonal antibody (A-6455, Invitrogen) and an Alexa Fluor® 488-conjugated anti-rabbit IgG secondary antibody (R37116, Invitrogen) at a dilution of 1:300 for both antibodies. The TUNEL assay was performed using an in situ cell death detection kit (Roche Applied Science, Mannheim, Germany). The sections were mounted using ProLong® Gold Antifade Mountant with DAPI (Thermo Fisher Scientific Inc.). Quantification was performed by separately counting TUNEL-positive cells (red) and total cells (blue) using ImageJ software.

**Antitumour activity in subcutaneous pancreatic tumour model.** BxPC3 cells were subcutaneously injected at a concentration of $1 \times 10^8$ cells per mouse into the right rear flank of each BALB/c nude mouse. The tumours were allowed to grow for 20 days before sample injection (or 4 days in the case of treatment with siVEGF). siLuc/uPIC or sihPLK1/uPIC (A/P = 10) was intravenously injected into the tail vein of each tumour-bearing mouse on the designated days (24 μg siRNA per mouse per shot), as represented by the arrows in Fig. 4i, j. The tumour volume and body weight of each mouse were measured for 8 weeks. The tumour volume ($V$) was calculated based on equation 7.

$$V = a \times b \times h \times \pi/6 \qquad (7)$$

where $a$, $b$, and $h$ are the major axis, minor axis, and height of the tumour, respectively, as measured using electronic callipers.

**Gene silencing activity in spontaneous pancreatic tumour.** The spontaneous pancreatic tumour model was developed in elastase 1-luciferase/SV40 T antigen transgenic mice (oncomice, Caliper Life Science, Hopkinton, MA, USA), in which the expression of luciferase is promoted specifically in acinar cell carcinoma[29]. uPIC solutions (A/P = 10) were intravenously injected into 13-week-old male oncomice (1.2 mg kg$^{-1}$ siLuc or siScr) without detectable tumour metastases. Bioluminescence intensity in the pancreatic tumour was determined before and 24 h after injection of uPIC using an IVIS instrument. Prior to the measurement, the mice were anesthetised with isoflurane, and luciferin was injected intraperitoneally at a dosage of 150 mg kg$^{-1}$. Ten minutes after luciferin injection, three different positions in each mouse (right flank, ventral, and left flank positions) displayed reduced bioluminescence variability owing to the tumour position. Photons emitted from the pancreas region were quantified using Living Image software and averaged from the three positions.

**Antitumour activity in spontaneous pancreatic tumour model.** Oncomice (13 weeks old, male) with similar tumour-derived bioluminescence intensities (~$2 \times 10^6$ photon s$^{-1}$ from the IVIS measurements) were randomly divided into two groups ($n = 10$). These mice were intravenously administered simPLK1/uPIC or siScr/uPIC (A/P = 10, 1.2 mg kg$^{-1}$ siRNA per shot) five times per week for the first 2 weeks and thereafter thrice per week. Liver metastasis in the oncomice was judged by the appearance of additional bioluminescence-positive positions in the IVIS images of the oncomice compared with the initial images.

**Blood counts and laboratory data.** uPIC (A/P = 10) was intravenously injected into the tail vein of each BALB/c mouse twice per week (24 μg siRNA per mouse per shot, a total of five shots). Mouse blood was collected 48 h after the final administration. Blood counts were analysed using an XT-2000iV (Sysmex, Kobe, Japan). Blood laboratory data were measured using a DRI-CHEM 7000i (Fujifilm) or analysed by SRL Medisearch (Tokyo, Japan).

**ASO/uPIC preparation and characterisation.** To prepare the ASO/uPIC, YBC and ASO were separately dissolved in 10 mM HEPES (pH 7.3) to produce concentrations of 500 μM and 100 μM, respectively. The two solutions were simply mixed by pipetting at the designated mixing ratios (A/P ratios) (final ASO concentration: 15 μM) with 150 mM NaCl and then incubated for 30 min at ambient temperature. For agarose gel retardation analysis, uPIC samples were prepared from A647-ASO at varying A/P ratios, mixed with glycerol, and applied to a 1% agarose gel. After electrophoresis (135 V, 10 min) in 1 × TBE running buffer, A647-ASO-derived bands were visualised using a Typhoon 9410 image scanner (GE Healthcare Life Science). For FCS analysis (or size/association number measurements), uPIC samples were prepared from YBC and A647-ASO at varying A/P ratios and diluted to 10 nM ASO with 10 mM HEPES buffer (pH 7.4) containing 150 mM NaCl. The autocorrelation curves of the fluorescent molecules obtained from 10 measurements at a sampling time of 10 s were fitted using the Zeiss Confocor3 software package to calculate the $D_C$ of the A647-ASO/uPIC. The obtained $D_C$ values were further converted to $D_H$ as described above. For AUC analysis, ASO/uPIC samples prepared at A/P = 1 and 20 were diluted to 1 μM ASO with 10 mM HEPES buffer (pH 7.4) containing 150 mM NaCl and subjected to measurements similar to those used for siRNA/uPIC. The PSV$_{ASO}$ was determined to be 0.431 using an AUC[39]. The PSV$_{ASO/PIC}$ was calculated as described above. For FRET analysis (or validation of dynamic ion-pairing, rendezvous), A594-YBC (300 μM, 100 μL), A647-ASO (30 μM, 100 μL), and N-YBC (300 μM, 100 μL) were sequentially administered through a preset catheter into the same mouse at $t = 1$, 6,

and 11 min, respectively, followed by the aforementioned observation using IVCLSM. For the blood circulation test, naked A647-ASO or A647-ASO/uPIC (A/P = 20) was intravenously injected into each mouse at 24 μg A647-ASO per mouse, followed by the aforementioned IVCLSM observation of the earlobe dermis.

**ASO delivery to orthotopic brain tumours.** To fabricate an orthotopic brain tumour model, glioma stem cells were harvested from patients (adult glioblastoma, grade IV, subtype: proneuronal, *EGFR*: amplified, *PTEN*: loss) who had undergone surgical treatment at Nagoya University Hospital (Nagoya, Japan)[11]. The patients provided written informed consent. The cells were routinely cultured in Neurobasal Medium (Life Technologies, Carlsbad, CA, USA) containing N2 and B27 supplements (Life Technologies), along with human recombinant basic fibroblast growth factor and epidermal growth factor (20 ng mL$^{-1}$ for each, R&D Systems, Minneapolis, MN, USA). Six-week-old female NOD/SCID mice (SLC, Shizuoka, Japan) were intracranially inoculated with the glioma stem cells[32] and fostered for 4 weeks before administration of ASO or ASO/uPIC. For the tumour accumulation study, naked A647-ASO or A647-ASO/uPIC (A/P = 20) was intravenously injected into the tumour-bearing mice (25 μg A647-ASO per mouse). At the designated time points after injection, the A647 fluorescence in the mice was recorded using an IVIS instrument. For determination of gene expression patterns by qRT-PCR, the tumour-bearing mice were intravenously administered asLuc/uPIC or asTUG1/uPIC (25 μg A647-ASO per mouse) three times at 24-h intervals. At 24 h after the last injection, whole brain was excised from each mouse, fixed in 4% PFA for 24 h, and washed in PBS. To confirm tumour formation, the fixed tumour tissue was embedded in paraffin, sectioned, and stained with HE. Total RNA was isolated from the tissue using TRIzol (Life Technologies), and an aliquot (1 mg) was used for reverse transcription with a SuperScript VILO cDNA Synthesis Kit (Life Technologies). Target gene (*TUG1*, *SOX2*, *MYC*, *BDNF*, *NGF*, and *NTF3*) expression levels were determined by TaqMan PCR (Applied Biosystems) or SYBR Green quantitative PCR based on the delta Ct method and then normalised to the housekeeping gene *GAPDH*[32]. To evaluate the antitumour activity of ASO/uPIC, the tumour-bearing mice were treated with asLuc/uPIC or asTUG1/uPIC (25 μg A647-ASO per mouse per shot) every 3 days for 4 weeks. Whole brain was then excised from each mouse, fixed in 4% PFA for 24 h, and washed in PBS. The fixed tumour tissues were embedded in paraffin, sectioned, and stained with HE. Images were obtained using a Leica DMI6000B microscope (Leica Microsystems, Wetzlar, Germany). Tumour volumes were calculated using the following formula: width ($W$) × height ($H$) × length ($L$)/2.

**Statistical analysis.** A two-sided Student's $t$-test and a Mann–Whitney $U$ test were used for single comparisons, in which a Gaussian distribution was assumed and not assumed, respectively, in the biodistribution study (Fig. 3g, Supplementary Fig. 10), the cellular uptake assay (Fig. 3n), the qRT-PCR assay (Fig. 4a), the quantitative western blot analysis (Fig. 4c), the gene silencing activity assay in the oncomice (Fig. 4k), the gene silencing assay in the orthotopic brain tumour (Fig. 5b), the orthotopic brain tumour volume (Fig. 5d), and the expression profiles of the related genes in the orthotopic brain tumour tissues (Supplementary Fig. 19). One-way analysis of variance (ANOVA) followed by the Tukey's test was used for the quantitative TUNEL assay (Fig. 4h). Two-way ANOVA was used for multiple comparisons in the antitumour activity assay for the subcutaneous pancreatic tumour model and mouse body weight measurements (Fig. 4i, j, and Supplementary Fig. 12). The Mantel-Cox test was used for comparison of survival curves of oncomice treated with uPICs (Fig. 4l). All obtained data were used for the statistical analyses without exclusion. The data were considered to be significantly different at $P < 0.05$.

**Others.** No human subjects and data depositions were used in this study.

**Reporting summary.** Further information on research design is available in the Nature Research Reporting Summary linked to this article.

## Data availability
The data that support the findings reported herein are available on reasonable request from the corresponding authors.

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

## Acknowledgements

This research was financially supported by the Funding Program for World-Leading Innovative R&D in Science and Technology (FIRST, JSPS), Grants-in-Aid for Scientific Research of MEXT (JSPS KAKENHI Grant Numbers 24659411 to S.W., 25000006 to K.K., 25282141, and 17H02098 to K.M.), the Center of Innovation (COI) Program (JST), the Project for Development of Innovative Research on Cancer Therapeutics (P-DIRECT, AMED), the Project for Cancer Research and Therapeutic Evolution (P-CREATE, AMED), the Basic Science and Platform Technology Program for Innovative Biological Medicine (IBIOMED, AMED), the Cooperative Research Program of "Network Joint Research Center for Materials and Devices: Dynamic Alliance for Open Innovation Bridging Human, Environment and Materials", and a grant JSPS Core-to-Core Program, A. Advanced Research Networks. The authors are deeply grateful to Professor K. Tsumoto (UTokyo) for his support in the density measurements. The authors are also grateful to Ms. A. Miyoshi (UTokyo) for her technical assistance.

## Author contributions

S.W. performed the in vivo experiments using subcutaneous pancreatic cancer model mice. K.H. performed the major physicochemical characterisations. K.T. performed the in vivo experiments by IVCLSM. H.J.K. performed the cellular uptake study in spheroid culture, synthesis of PEG-siRNA and its related experiments, and western blotting. X.L. performed the in vivo experiments using oncomice. H.Ch. contributed to several physicochemical characterisations and the animation production. S.F. contributed to the polymer synthesis. Ke.K. and Y.K. contributed to the in vivo experiment using orthotopic brain tumour model mice. Sa.U., S.O., T.No., and H.T. contributed to several in vivo experiments. H.Ca., H.K., H.Y.T., M.R.K., Y.M., H.F., Sh.U., and M.N. provided advice on the design of animal experiments. K.O. and N.N. provided advice on the design of whole experiments. K.M. and Ka.K. designed the research concept, managed the project, and were the main contributors to the manuscript writing.

## Additional information

**Competing interests:** N.N., K.M., and K.Kata. are scientific advisors of AccuRna, Inc. The remaining authors declare no competing interests.

