## [Peer Review File · Nature Communications]

Reviewers' comments:

Reviewer #1 (Remarks to the Author):

In comments to the editor the reviewer said they were happy with the revisions.

Reviewer #3 (Remarks to the Author):

The authors have thoroughly addressed my concerns. It is an exciting advance in delivery methods for nucleic acid therapeutics.

Reviewer #5 (Remarks to the Author):

The authors have revised their manuscript, and improved various descriptive and experimental aspects. They better describe the model systems used, which is important for the readers. Overall, this paper is about a new technical approach and not new biology, and as such it is a contribution for the field to consider.

Reviewer #6 (Remarks to the Author):

Reviewer 4 requested a head to head comparison of the siRNA-uPIC therapy and a standard chemotherapy, presumably so that the reader might be able to (1) better interpret the survival data presented in this manuscript and (2) more clearly understand the 'delivery barriers' to conventional treatments in these tumor models. In my opinion, the authors partially, but satisfactorily, respond to the first point but fall short of adequately addressing the second point.

-The authors now cite survival data (from their own prior work) for the efficacy of platinum-based chemotherapy in the spontaneous tumor model. Traditional chemotherapy in the cited paper showed essentially no benefit, although a polymeric micelle-Pt nanodrug enhanced survival considerably. (Apparently polymeric nanoparticles do work reasonably well in this model, perhaps weakening the argument that smaller uPICs are needed to overcome delivery barriers.) Nonetheless, the cited literature partially addresses the question of efficacy of siRNA-uPIC vs chemotherapy and provides some context for interpreting survival curves for at least one of the models.

-The authors further explain in their response that the antisense molecules and targets will be optimized in a future study, which I found to be satisfactory. To my mind, the potential advances in this manuscript deal with the delivery vehicle rather than the specific cargo being delivered or superior tumor killing. Comparison to chemotherapy seems less important (for now) than the need for rigorous comparisons of circulation, penetration, uptake, and target knockdown between uPICs and other appropriate siRNA delivery modalities (discussed below).

-This relates to Reviewer 4's comment that the 'molecular design is similar to many past approaches'. I again agree with the authors' response that their design is indeed different, but unfortunately they haven't performed experiments to directly demonstrate that their design leads to improved performance. The authors chose as controls (i) naked siRNA, which is fine to show but there is little to reason to expect would be successful, and (ii) a commercial lipid product InVivofectamine, which I'm not convinced is the most appropriate. There are several issues here. While such lipid nanoparticles are indeed larger than uPICs, they also have different (presumably proprietary) surface chemistry and are marketed primarily for knockdown of liver proteins. A more appropriate siRNA delivery vehicle comparison might be the micellar PICs with a PEG corona depicted in Fig S2. The authors spend the first part of the manuscript arguing the potential advantages (e.g. size, dynamic equilibrium) of their

unit PICs versus micellar PICs, but there is no comparison of these materials in vitro or in vivo.

-Reviewer 4 requested the comparison to chemotherapy for a 'more rigorous description of delivery barriers'. I don't believe that chemotherapy is necessary for this, but questions do remain concerning the suitability of the models and the nature of 'delivery barriers' that cause other therapies to fail. Most notably, accumulation of uPICs in tumors is apparently dependent on the highly variable and model-specific EPR effect. The relevance of EPR to 'intractable' human cancers such as hypovascular, desmoplastic PDAC remains unclear. The notion that the authors' material might overcome a fibrotic delivery barrier in model tumors must be weighed against the acknowledgment that they also rely on features of the model that clearly promote delivery and accumulation, and that these features might not be characteristics of intractable human cancers. Similarly, the treatment of the glioma model is impressive, but a lack of characterization of the 'delivery barrier' here is also concerning, particularly the fact that the authors can only state at the end of the manuscript that it '**may**' have featured a blood brain tumor barrier'.

Reviewer #1

In comments to the editor the reviewer said they were happy with the revisions.

Answer: We really appreciate your efforts and positive evaluation of our revised manuscript.

Reviewer #3

The authors have thoroughly addressed my concerns. It is an exciting advance in delivery methods for nucleic acid therapeutics.

Answer: We really appreciate your efforts and positive evaluation of our revised manuscript.

Reviewer #5

The authors have revised their manuscript, and improved various descriptive and experimental aspects. They better describe the model systems used, which is important for the readers. Overall, this paper is about a new technical approach and not new biology, and as such it is a contribution for the field to consider.

Answer: We really appreciate your efforts and positive evaluation of our revised manuscript.

Reviewer #6

Generals: Reviewer 4 requested a head to head comparison of the siRNA-uPIC therapy and a standard chemotherapy, presumably so that the reader might be able to (1) better interpret the survival data presented in this manuscript and (2) more clearly understand the ‘delivery barriers’ to conventional treatments in these models. In my opinion, the authors partially, but satisfactorily, respond to the first point but fall short of adequately addressing the second point.

Answer: Thank you for reviewing our revised manuscript on behalf of the reviewer #4. As you suggested, we have worked on more clearly describing the delivery barriers to conventional nanomedicines, which was particularly summarized in our answer to your 4th comment, as below.

1) The authors now cite survival data (from their own prior work) for the efficacy of platinum-based chemotherapy in the spontaneous tumor model. Traditional chemotherapy in the cited paper showed essentially no benefit, although a polymeric micelle-Pt nanodrug enhanced survival considerably. (Apparently polymeric nanoparticles do work reasonably well in this model, perhaps weakening the argument that smaller uPICs are needed to overcome delivery barriers.) Nonetheless, the cited literature partially addresses the question of efficacy of siRNA-uPIC vs chemotherapy and provides some context for interpreting survival curves for at least one of the models.

Answer: Thank you for reviewing not only the present study but also our previous study [*PNAS* 110, 11397–11402 (2013)]. One thing that we would like to explain with supplements is the size of the platinum drug (Pt)-loaded polymeric micelle used in the previous study. Its hydrodynamic diameter was 30 nm, which was optimized in comparison with 50, 70, and 100 nm-sized counterparts for enhanced penetrability into the pancreatic cancer tissues in another previous study [*Nat. Nanotechnol.* 6, 815–823 (2011)]. Thus, the high anticancer efficacy of the Pt-loaded polymeric micelle with 30 nm in size matches well with our argument that smaller delivery vehicles are needed to overcome the delivery barrier (or thick fibrotic stromal tissues with hypovascularity). It should be also noted that the sizes of conventional siRNA-loaded polymeric micelles were ~40 nm or larger in our previous studies, e.g., *ACS Nano* 6, 5174–5189 (2012); *Biomaterials* 35, 7887–7895 (2014). It was substantially difficult to further downsize the siRNA-loaded micelles, presumably due to the rigid cylindrical architecture of double-stranded RNA molecules (length: ~6 nm, width: ~2 nm). Alternatively, the uPIC formation presented here has an advantage for the fabrication of smaller siRNA vehicles with a size of less than 30 nm.

2) The authors further explain in their response that the antisense molecules and targets will be optimized in a future study, which I found to be satisfactory. To my mind, the potential advances in

this manuscript deal with the delivery vehicle rather than the specific cargo being delivered or superior tumor killing. Comparison to chemotherapy seems less important (for now) than the need for rigorous comparisons of circulation, penetration, uptake, and target knockdown between uPICs and other appropriate siRNA delivery modalities (discussed below).

Answer: Thank you for understanding the important point of the present study, *i.e.*, dealing with the delivery vehicle rather than the specific cargo. We agree to your comment that the more important is the characterizations of uPIC in comparison with other delivery modalities. This point was described with more details in our answer to your 3rd comment, as below.

3) This relates to Reviewer 4's comment that the 'molecular design is similar to many past approaches'. I again agree with the authors' response that their design is indeed different, but unfortunately they haven't performed experiments to directly demonstrate that their design leads to improved performance. The authors chose as controls (i) naked siRNA, which is fine to show but there is little to reason to expect would be successful, and (ii) a commercial lipid product InVivofectamine, which I'm not convinced is the most appropriate. There are several issues here. While such lipid nanoparticles are indeed larger than uPICs, they also have different (presumably proprietary) surface chemistry and are marketed primarily for knockdown of liver proteins. A more appropriate siRNA delivery vehicle comparison might be the micellar PICs with a PEG corona depicted in Fig S2. The authors spend the first part of the manuscript arguing the potential advantages (e.g. size, dynamic equilibrium) of their unit PICs versus micellar PICs, but there is no comparison of these materials in vitro or in vivo.

Answer: Thank you for the important comment that uPIC should be appropriately compared with other relevant controls. As you mentioned, InVivofectamine LNP has a different material composition with a larger size and its recommended target is not cancer but liver. Nevertheless, the reason for employing InVivofectamine as a control is that we acknowledged the previous suggestion from the reviewer #2. He/she suggested an LNP as a '**practical**' control, and accordingly, we added the data obtained by the use of InVivofectamine as a practical control. We see your point that a more relevant control may be micellar PICs, which are secondary associates of uPICs as illustrated in Supplementary Figure 2. However, YBC (2-armed PEG-*b*-polylysine) did not undergo secondary association to form siRNA-loaded micellar PICs because of the rigid cylindrical architecture of siRNA [ref 12] as well as the large steric hindrance of long 2-armed PEG (2 × 37 kDa). To fabricate micellar PICs, it may need substantial modifications in the structure of block copolymers, e.g., use of shorter single-armed PEG to reduce steric repulsion as well as introduction of hydrophobic moieties into the polylysine side chain to increase the cohesive force between uPICs. These substantial modifications apparently result in the change in the inherent

characteristics of whole carrier systems, and eventually, the results cannot be simply argued by the size effect. Alternatively, we selected a PEGylated siRNA conjugate (PEG-siRNA) as a control having a '**similar size and composition**' to uPIC developed here (see Supplementary Figures 8 and 9). Actually, use of PEG-siRNA as a control is also the suggestion from the previous reviewer #1, and we acknowledged this suggestion to compare PEG-siRNA with uPIC. Indeed, we prepared the PEG-siRNA by copper-free click conjugation between 2-armed PEG-N₃ and DBCO-functionalized siRNA (Supplementary Figure 8a) and purification by gel filtration chromatography (Supplementary Figure 8b). Nevertheless, in a sharp contrast with high RNase-tolerability of uPIC, the obtained PEG-siRNA was readily degraded in RNase A-containing media (Supplementary Figure 9), and eventually, not worthy for *in vivo* studies as described in the text (page 7, line 4–13). This result clearly highlights an improved performance of dynamic equilibrium mechanism (Rendezvous mechanism) involved in uPIC system for siRNA delivery compared to covalent bonding system of PEG with siRNA.

4-1) Reviewer 4 requested the comparison to chemotherapy for a 'more rigorous description of delivery barriers'. I don't believe that chemotherapy is necessary for this, but questions do remain concerning the suitability of the models and the nature of 'delivery barriers' that cause other therapies to fail. Most notably, accumulation of uPICs in tumors is apparently dependent on the highly variable and model-specific EPR effect. The relevance of EPR to 'intractable' human cancers such as hypovascular, desmoplastic PDAC remains unclear. The notion that the authors' material might overcome a fibrotic delivery barrier in model tumors must be weighed against the acknowledgment that they also rely on features of the model that clearly promote delivery and accumulation, and that these features might not be characteristics of intractable human cancers.

Answer: Thank you for the thoughtful comment related to the delivery barriers in the tumor models and their relevance to intractable human cancers. As you mentioned, the accumulation of uPICs in tumors mainly relies on the so-called EPR effect and the extent of EPR effect (or the accumulation efficiency of nanomedicine) significantly varies between tumor models, presumably due to their varying histological characteristics, *i.e.*, cancer model heterogeneity, such as different vascular density and fibrotic tissue volume. Although the present pancreatic cancer model was carefully selected to meet a criterion of clinically relevant surrogate, as you critically pointed out, the effectiveness of uPIC formulation would need to be further verified against a wide variety of tumor models in future studies to justify the relevance in overcoming the delivery barriers (or histological characteristics) in clinical settings. Accordingly, we have modified several sentences in the revised manuscript to accommodate your critical comments as follows.

In page 7, line 16:

Original sentence: “Recent studies on cancer pathophysiology have shown that the nature of the tumour and its associated tissue significantly affects the accumulation and penetration profiles of compounds extravasating from the blood vascular compartment⁴⁻⁸. In particular, pancreatic cancer commonly involves a thick fibrotic stroma with hypovascularity and generates tumour microenvironments that are inaccessible to anticancer drugs and imaging agents^{5,6}, resulting in the lowest five-year survival rate of any cancers^{17,18}. This is a critical issue for cancer nanomedicine that relies on particulate carrier systems.”

Revised sentence: “Many previous studies have demonstrated that nanoparticle carrier systems can efficiently accumulate in solid tumour models through the leaky tumour vasculature and immature lymphatic drainage, termed enhanced permeability and retention (EPR) effect^{17,18}. In this regard, recent studies on cancer pathophysiology have revealed that the nature of the tumour and its associated tissue significantly affects the accumulation and penetration profiles of nanoparticles extravasating from the bloodstream⁴⁻⁸. In particular, pancreatic cancer commonly involves a thick fibrotic stroma with hypovascularity, generating tumour microenvironments that are inaccessible to conventional nanoparticle carrier systems with a size of ~100 nm^{5,6}. These characteristic tumour microenvironments are a current critical issue for cancer nanomedicine.”

In page 7, line 32:

Original sentence: “This high availability of uPIC is presumably due to its size of several tens nm or less and significant retention in the blood, allowing for enhanced extravasation from the leaky tumour vasculature and retention in the tumour tissue via so-called enhanced permeability and retention (EPR) effect²¹.”

Revised sentence: “This high availability of uPIC is apparently consistent with our previous results that small nanomedicine with a size of 30 nm can efficiently accumulate in the same tumour model, whereas counterparts above 50 nm did not show such accumulation⁶.”

In page 12, line 4:

Original sentence: “Thus, the uPIC holds great promise for the systemic use of oligonucleotide therapeutics against intractable cancers with tight access barriers.”

Revised sentence: “While the present SNA delivery strategy of uPIC needs to be further verified for a wide variety of tumour models to gain better insight into the relevance to clinical human cancers, it demonstrates the strong potential for the systemic use of oligonucleotide therapeutics against intractable cancers with tight access barriers.”

Reference:

17. Matsumura, Y. & Maeda, H. A new concept for macromolecular therapeutics in cancer chemotherapy: mechanism of tumorotropic accumulation of proteins and the antitumor agent smancs. *Cancer Res.* **46**, 6387–6392 (1986).
18. Maeda, H. Toward a full understanding of the EPR effect in primary and metastatic tumors as well as issues related to its heterogeneity. *Adv. Drug Deliv. Rev.* **91**, 3–6 (2015).

4-2) Similarly, the treatment of the glioma model is impressive, but a lack of characterization of the ‘delivery barrier’ here is also concerning, particularly the fact that the authors can only state at the end of the manuscript that it ‘**may** have featured a blood brain tumor barrier’.

Answer: Thank you for the comment related to the blood brain tumor barrier. We have noticed that the previous manuscript did not sufficiently explain about it. Thus, we have added the explanation on the blood brain tumor barrier in the revised manuscript as blow.

In page 10, line 26:

Original sentence: “In the present study, we tested the ASO/uPIC (A/P = 20) in an intracranial xenograft mouse model using glioma stem cells obtained from the surgical treatment of patients¹⁰. Brain tumours are widely known to be another intractable form of cancer that features the so-called blood-brain tumour barrier, which restricts the extravasation of nanoparticles from the bloodstream to the tumour tissue^{7,8,32}.”

Revised sentence: “Brain tumours are widely known to be another intractable form of cancer that features the so-called blood-brain tumour barrier or blood-tumour barrier. Although the barrier characteristics, *e.g.*, tight junction, narrow perivascular space, absence of fenestrae, and inactive pinocytosis, in the brain vasculature may be compromised in brain tumours³¹, they still act as the obstacles restricting the extravasation of nanoparticles from the bloodstream to the tumour tissue^{7,8}. In the present study, we challenged to treat an intracranial xenograft mouse model of glioma stem cells obtained from the surgical treatment of patients¹⁰ by the use of the ASO/uPIC (A/P = 20).

We really appreciate your careful reviewing of our manuscript and giving the insightful comments. We hope that the revised manuscript is satisfactorily corrected for publication in *Nature Communications*.

REVIEWERS' COMMENTS:

Reviewer #6 (Remarks to the Author):

The authors' responses and revisions were sufficient to clarify the delivery barriers and limitations in their tumor models.